# E3 ubiquitin ligase RNF10 promotes dissociation of stalled ribosomes and responds to ribosomal subunit imbalance

Janina A. Lehmann [1,2], Doris Lindner[1,2], Hsu-Min Sung[1,2] & Georg Stoecklin [1,2] ✉

Aberrant translation causes ribosome stalling, which leads to the ubiquitination of ribosomal proteins and induces ribosome-associated quality control. As part of this quality control process, the E3 ubiquitin ligase RNF10 monoubiquitinates ribosomal protein RPS3. Here, we demonstrate that RNF10-mediated RPS3 monoubiquitination antagonizes ribosomal half-mer formation by promoting dissociation of 40S subunits from ribosomes stalled during translation elongation. Interestingly, RNF10 also promotes dissociation of 40S subunits stalled during aberrant translation initiation. Moreover, RNF10 levels are tightly coupled to the amount of 40S subunits. Knockdown of RPS proteins, which abrogates 40S ribosome biogenesis, results in proteasomal degradation of RNF10. Vice versa, knockdown of RPL proteins, which abrogates 60S biogenesis, leads to the accumulation of stalled initiating 40S subunits, increased RNF10 levels, and RPS3 monoubiquitination. As a factor required for the resolution of stalled translation events, RNF10 is part of a fundamental mechanism by which cells respond to imbalances in ribosomal subunit stoichiometry.

Translation of mRNA into protein is tightly regulated and monitored by quality control processes to sustain cellular homeostasis[1,2]. Translational stress, defects in the translation machinery as well as errors in mRNAs can lead to stalling and collision of elongating ribosomes[3–6]. Ribosome stalling events occur under various conditions such as strong mRNA secondary structures, mRNAs lacking stop codons, fluctuations in tRNA levels, aberrant tRNA modifications, decoding of suboptimal codons, specific nascent chain sequences, chemical damage of mRNA or ribosomes, or the presence of immature ribosomal subunits[3,6–8]. While many of these events affect individual mRNAs only, pharmacological translation elongation inhibitors, also termed ribotoxins, induce wide-spread ribosome stalling and lead to a pronounced ribotoxic stress response[9,10]. Recent advances have established molecular details of how ribosome stalling activates the ribosome-associated quality control (RQC) pathway, which entails

ubiquitination followed by subsequent degradation of the nascent polypeptide chain, as well as ubiquitination of multiple small ribosomal proteins[11–13]. The latter ubiquitination events induce the splitting and dissociation of stalled ribosomal subunits[14–16], which are either degraded or recycled for further rounds of translation, and additionally promote degradation of the affected mRNA by the non-stop or no-go mRNA decay pathways[6,17,18]. Together, these mechanisms ensure efficient removal of potentially harmful RNA and protein products.

Mechanistically, ribosome stalling leads to the collision of two or more ribosomes, forming RQC complexes with a unique structural interface between the neighboring small ribosomal subunits of the initially stalled (leading) ribosome and the upstream, collided ribosome[19,20]. In mammalian RQC, the human zinc finger E3 ubiquitin ligase ZNF598 recognizes the interfaces of these collided 40S subunits and ubiquitinates ribosomal proteins RPS3, RPS10 and RPS20[13].

[1]Division of Biochemistry, Mannheim Institute for Innate Immunoscience (MI3) and Mannheim Cancer Center (MCC), Medical Faculty Mannheim, Heidelberg University, 68167 Mannheim, Germany. [2]Center for Molecular Biology of Heidelberg University (ZMBH), German Cancer Research Center (DKFZ)-ZMBH Alliance, 69120 Heidelberg, Germany. ✉e-mail: georg.stoecklin@medma.uni-heidelberg.de

Absence or mutation of ZNF598 results in defective ubiquitination, and failure of proper RQC activation[13,20]. In yeast, a two-step mechanism was described whereby the E3 ubiquitin ligase Mag2 first mono-ubiquitinates uS3 (RPS3 in humans), while yeast Hel2 (human ZNF598) and a further E3 ubiquitin ligase, Rsp5, polyubiquitinate uS3[21]. These events lead to dissociation of non-functional ribosomal subunits and to rapid 18S non-functional rRNA decay (NRD)[21]. The human homologue of Mag2, RING finger protein 10 (RNF10), was described in the nervous system as a transcription factor and regulator of Schwann cell differentiation and myelination[22], as a synaptonuclear messenger involved in neuronal morphology[23], and as a gene associated with adiposity in mice and humans[24,25]. In the context of RQC, RNF10 was found to monoubiquitinate RPS3 upon both translation initiation and elongation stalling, and thereby promote the turnover of stalled 40S subunits[26,27]. By reconstitution of stalled ribosome complexes, ubiquitination of RPS3 by RNF10 was recently also confirmed in vitro[28].

In the study presented here, we show that RNF10 promotes the dissociation of stalled 40S subunits from the mRNA, and that RNF10 levels are tightly coupled to the stoichiometric balance between 40S and 60S subunits. Our results point towards an important function of RNF10 in the resolution of stalled translation events, providing a link between translational stress and the control of ribosomal subunit stoichiometry.

## Results

### RNF10 monoubiquitinates ribosomal protein RPS3 within stalled 40S subunits

Upon ribosome stalling, a variety of ubiquitination events on small ribosomal proteins were shown to contribute to the proper initiation and progression of RQC[13,29]. Here, we aimed to further understand the role of the human E3 ubiquitin ligase RNF10 in the context of ribosome stalling. We first silenced RNF10 expression with siRNAs in human cervix carcinoma HeLa cells and monitored RPS3 as a known substrate by Western blot analysis (Fig. 1a, quantification in Supplementary Fig. 1a). In the control knockdown (KD), RPS3 is monoubiquitinated (Ub$_1$-RPS3) when cells are treated with the translation elongation inhibitor anisomycin (ANI), while RPS3 monoubiquitination is strongly diminished upon KD of RNF10 (siRNA #214; Fig. 1a), as observed previously[26,27]. The same effect was seen with a second siRNA (#213) against RNF10 (Supplementary Fig. 1b) as well as with two additional translation elongation inhibitors, cycloheximide (CHX) and blasticidine (BLA), which induce ribosome stalling by binding to different sites on the ribosome[30] (Fig. 1a, Supplementary Fig. 1b). Likewise, we found that RNF10 mediates RPS3 monoubiquitination upon ANI treatment in RPE1 cells (Supplementary Fig. 1c and d), a non-cancerous, immortalized human retinal pigment epithelial cell line.

To ascertain this finding, we generated HeLa cells with a knockout (KO) of RNF10 by CRISPR/Cas9-mediated targeting of *RNF10* exon 5 (encoding the C3HC4 ring finger motif, locus 29) or exon 2 (locus 81; Supplementary Fig. 1e). Several RNF10-KO clones were obtained, all of which failed to monoubiquitinate RPS3 upon ANI treatment (Fig. 1b, Supplementary Fig. 1f), corroborating the result above. To avoid clonal effects, five RNF10 KO locus 29 clones and five locus 81 clones were pooled for some of the experiments, and absence of RPS3 monoubiquitination was confirmed in these pools (Supplementary Fig. 1g). These results consolidate previous evidence that RNF10 is the primary E3 ubiquitin ligase responsible for RPS3 monoubiquitination upon translation elongation stalling[26,27].

The E3 ligase ZNF598 was also reported to contribute to RPS3 ubiquitination upon ribosome stalling in HCT116 and HEK293T cells[13], yet KD of ZNF598 in our hands had no consistent effect on RPS3 monoubiquitination in HeLa cells subjected to the translation elongation inhibitors ANI, CHX or BLA (Fig. 1a, Supplementary Fig. 1h, i). Interestingly, KD of ZNF598 led to elevated expression of RNF10 in

RPE1 cells (Supplementary Fig. 1c and d), an effect that was, however, not observed in HeLa cells (Fig. 1a, Supplementary Fig. 1a and i).

In addition to its structural role within the 40S ribosomal subunit, RPS3 has several extraribosomal functions[31], e.g. in mitochondrial DNA damage surveillance[32], in microbial pathogenesis[33], or as a factor associated with NF-κB mediated transcriptional control[34]. To test whether RNF10 monoubiquitinates RPS3 within or outside of the ribosome, we performed Western blot analysis following polysome fractionation. Given that EDF1 is a collision sensor that binds to stalled collided di-ribosome complexes[35], its accumulation in polysomal fractions upon ANI treatment confirmed translation elongation stalling (Fig. 1c). In parental and control KO HeLa cells treated with ANI, Ub$_1$-RPS3 was observed in the 40S and 80S fractions, and, to some degree, also in polysomal fractions (Fig. 1c, quantification in Fig. 1d). In contrast, little monoubiquitination was observed in the pool of free RPS3 (fraction #1). As expected, ubiquitinated RPS3 was not detected in any of the fractions from the RNF10-KO pools (Fig. 1c). These results argue that RNF10 monoubiquitinates RPS3 within 40S ribosomal subunits during translation elongation stalling.

### RNF10 suppresses formation of stalled ribosomal half-mers

To further explore the function of RNF10 as an E3 ubiquitin ligase, we used a rescue approach whereby HA-tagged RNF10-wild type (WT) was stably reintroduced into HeLa RNF10-KO cells through lentiviral transduction (HeLa-KO+RNF10-WT). We also expressed an HA-tagged RNF10-C225S/C228S mutant (Mut), wherein the E3 ubiquitin ligase activity was disrupted through mutation of the first two cysteines of the C3HC4 RING finger domain (HeLa-KO+RNF10-Mut). In HeLa-KO+RNF10-WT cells, RPS3 was weakly monoubiquitinated already under basal conditions, and its monoubiquitination became stronger than in parental HeLa cells upon ANI treatment, in line with the higher expression level of RNF10 in these cells (Fig. 2a). In contrast, Ub$_1$-RPS3 was not detected in HeLa-KO+RNF10-Mut cells, as in RNF10-KO cells (Fig. 2a), consistent with the effect of a similar RNF10 mutant published previously[26]. Hence, RPS3 monoubiquitination is directly dependent on the level and E3 ligase activity of RNF10.

Next, we explored whether RNF10 activity influences translation by conducting polysome profile analysis. While the profiles of RNF10-KO cells were barely different from parental HeLa cells, elevated expression of RNF10-WT led to a strong, more than 2-fold decrease of free 40S subunits, and a corresponding increase of free 60S subunits (red curves in Fig. 2b, quantification in Supplementary Fig. 2a). As previously observed in HEK293 cells[26], this indicates that RNF10 promotes the clearance of stalled RQC complexes, which leads to the degradation of 40S particles and an accumulation of free 60S particles.

An additional effect was observed when cells were exposed to ANI, as it led to the pronounced formation of ribosomal half-mers in parental HeLa cells (yellow inset in Fig. 2b). Since we used ANI at an intermediate dose of 0.1 μg/ml, it inhibits elongation of only a subset of ribosomes and thereby efficiently induces ribosome collisions (whereas collisions are less frequent when all ribosomes are stalled at higher doses of ANI). In yeast, it was shown that ribosome collisions are cleared in a multi-step process that involves 60S subunit dissociation from the stalled lead ribosome and formation of 40S-containing ribosomal half-mer intermediates[14]. Within such half-mers, the remaining 40S subunit is still connected to the upstream collided ribosome via its mRNA association. As expected from the yeast data, we observed half-mer formation in HeLa cells upon treatment with ANI at an intermediate, but not at a high dose (Supplementary Fig. 2b). Importantly, the formation of polysomal half-mer intermediates was suppressed in HeLa cells expressing elevated levels of RNF10-WT, yet unaffected in RNF10-Mut expressing cells (Fig. 2b). Likewise, ribosomal half-mer formation was suppressed by elevated levels of RNF10-WT when cells were treated with an intermediate dose of CHX (Supplementary Fig. 2c, d).

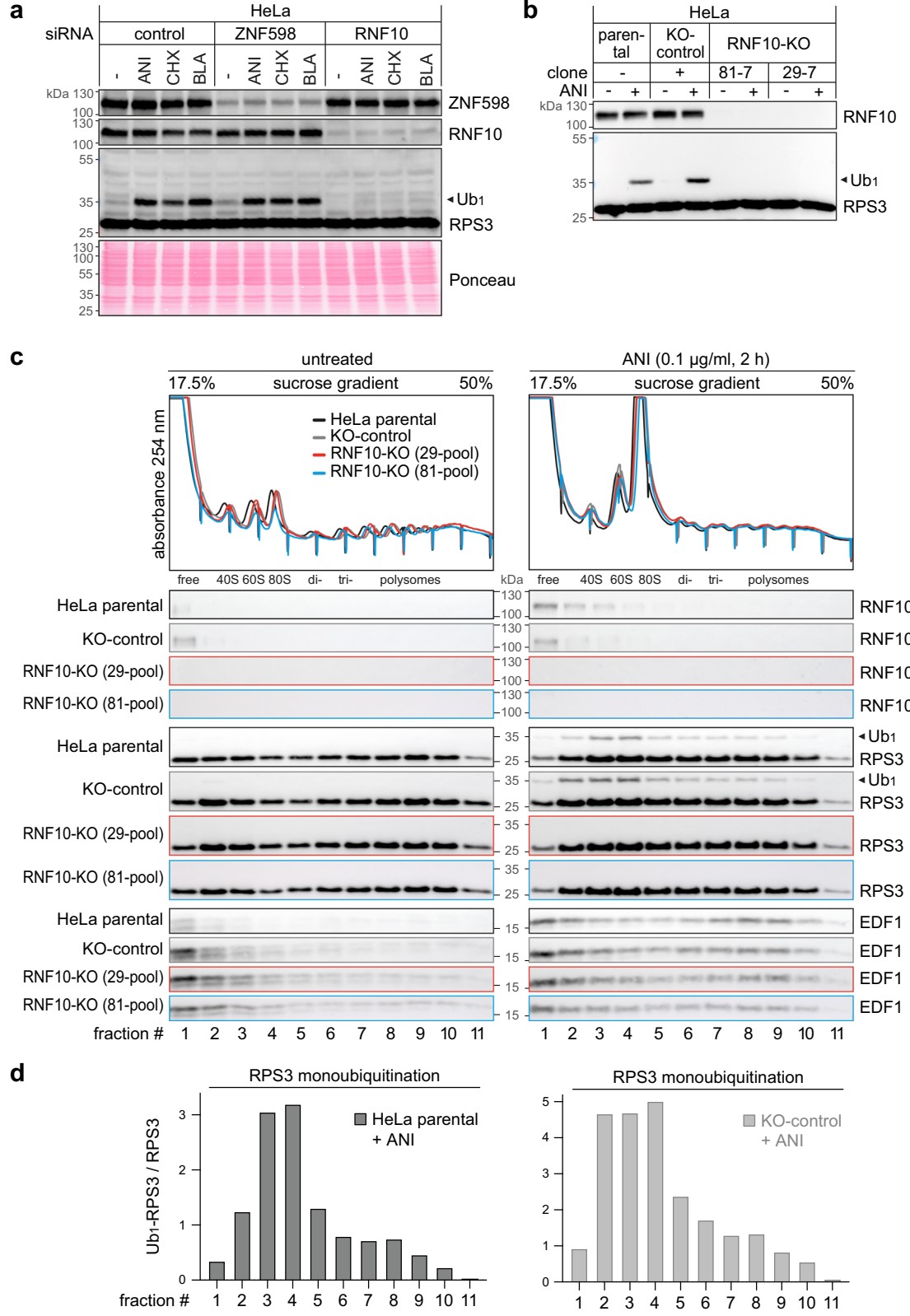

**Fig. 1 | RPS3 monoubiquitination by RNF10 upon translation elongation stalling. a** Following control KD, KD of ZNF598 (S193) or KD of RNF10 (S214) for 72 h, HeLa cells were treated for 2 h with translation elongation inhibitors anisomycin (ANI, 0.1 μg/ml), cycloheximide (CHX, 100 μg/ml), or blasticidine (BLA, 100 μg/ml). RPS3 monoubiquitination (Ub1) was assessed by Western blot analysis. **b** Western blot analysis showing the effect of ANI treatment (2 h, 0.1 μg/ml) on Ub1-RPS3 in RNF10-KO HeLa cells (clones 81-7 and 29-7) compared to parental HeLa cells or a KO-control clone. The blot is representative of 3 independent experiments. **c** Polysome profile and fractionation analysis of HeLa RNF10-KO cell pools; the distribution of RNF10 and Ub1-RPS3 in the absence and presence of ANI treatment (2 h, 0.1 μg/ml) was assessed by Western blot analysis. Sharp peaks pointing downwards are artefacts of the electric signal of the fractionator. **d** The Ub1-RPS3 / RPS3 ratio ($n = 1$ biological replicate) in parental and KO-control HeLa cells in the presence of ANI was calculated by quantification of Western blots in panel (**c**). Source data are provided as a Source Data file.

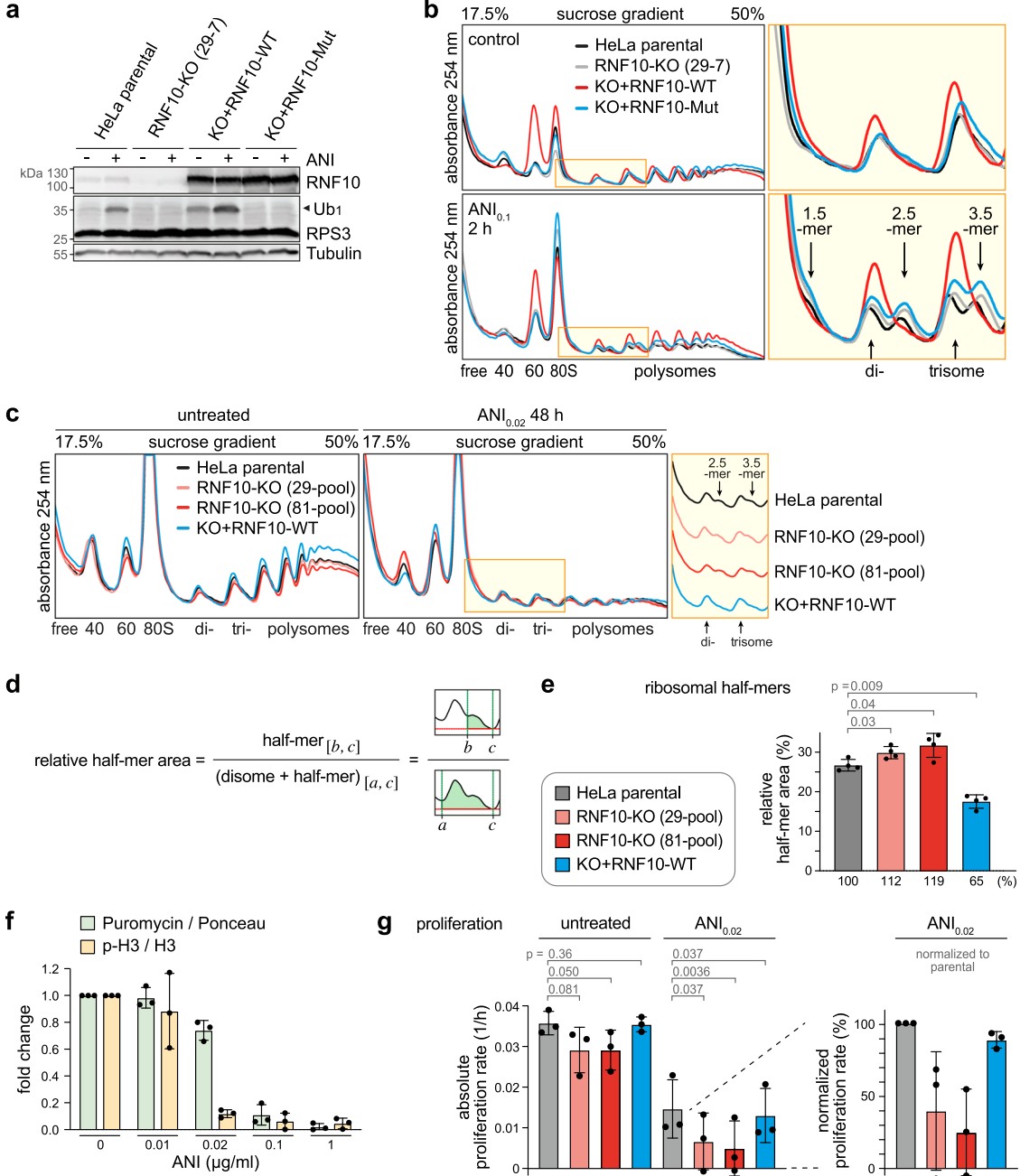

**Fig. 2 | Suppression of ribosomal half-mer formation by RNF10. a** Western blot analysis of Ub₁-RPS3 in HeLa-RNF10-KO cells stably expressing HA-RNF10-WT (KO +RNF10-WT) or HA-RNF10-Mut (C225S/C228S; KO+RNF10-Mut), ± intermediate dose ANI treatment (2 h, 0.1 μg/ml); Tubulin serves as loading control. The blot is representative of 2 independent experiments. **b** The same cells as in panel (**a**) were subjected to polysome profile analysis; the yellow area is enlarged on the right side. **c** Polysome profiles of parental HeLa cells, HeLa-RNF10-KO pools and HeLa-KO +RNF10-WT cells, ± low dose ANI treatment (48 h, 0.02 μg/ml); profiles are shown separately in the yellow area. **d** Formula and illustration for the quantification of ribosomal half-mers; half-mer area [b, c] and (disome + half-mer) area [a, c] are shown in green; baseline of the polysome profile in red. **e** Ribosomal half-mers (2.5 mer / disome ratio) from polysome profiles as in panel (**c**) were quantified

according to panel (**d**), shown are mean values ± SD (n = 4 biological replicates; p-values determined by two-tailed, paired t-test). **f** HeLa cells were exposed to increasing concentrations of ANI for 2 h, followed by measurement of total protein synthesis by puromycin incorporation, and assessment of proliferation by monitoring phosphorylation of histone H3 at serine 10. Shown are mean values ± SD (n = 3 biological replicates) based on the quantification of Western blots as in (Supplementary Fig. 2g). **g** Absolute proliferation rate (left) and relative proliferation rate (right) of parental HeLa cells, HeLa-RNF10-KO pools and HeLa-KO+RNF10-WT cells, ± low dose ANI treatment (72 h, 0.02 μg/ml); color code as in panel (**e**). Shown are mean values ± SD (n = 3 biological replicates; p-values determined by one-tailed, paired t-test). Source data are provided as a Source Data file.

We then used a low dose of ANI (0.02 μg/ml), which allowed us to culture cells over extended periods of time and assess their response to chronic translational stress. After 48 h in presence of low dose ANI, loss of half-mer formation was again obvious in cells overexpressing RNF10-WT (Fig. 2c). Importantly,

careful quantification of half-mers (Fig. 2d) with help of the QuAPPro application[36] revealed elevated half-mer formation in the two RNF10-KO pools (Fig. 2e). While the difference compared to parental HeLa cells was small, it was statistically significant (Fig. 2e).

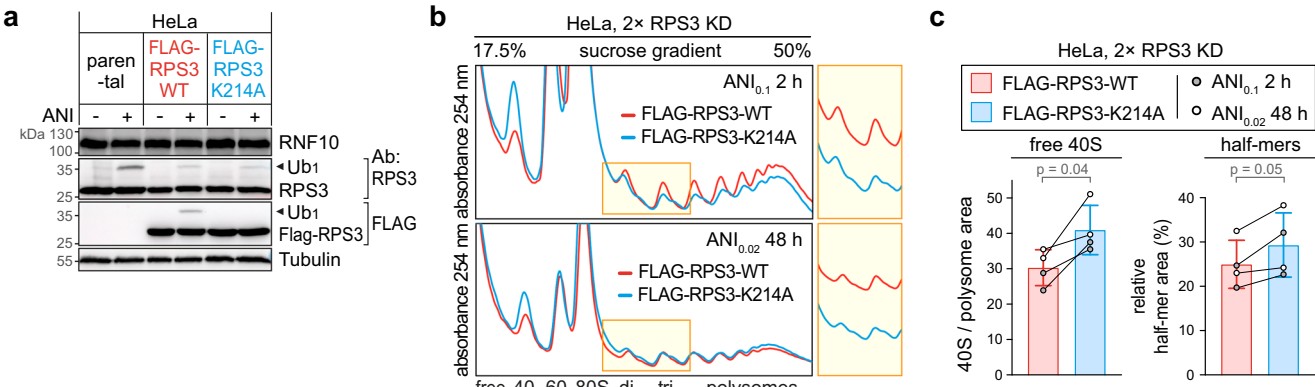

**Fig. 3 | RPS3 ubiquitination at K214 promotes dissociation of stalled 40S subunits. a** FLAG-RPS3-WT and FLAG-RPS3-K214A were stably expressed in HeLa cells, and $Ub_1$-RPS3, ± ANI treatment (2 h, 0.1 μg/ml), was assessed by Western blot analysis; Tubulin serves as loading control. The blot is representative of 3 independent experiments. **b** Polysome profile analysis of cells as in panel (**a**) after KD of RPS3 (siRNA S292) on day 1 followed by a second KD of RPS3 on day 3. Polysome profiles were recorded on day 5 after intermediate-dose ANI treatment (2 h, 0.1 μg/ml) or on day 6 after low-dose ANI treatment (for 48 h from day 4–6, 0.02 μg/ml). **c** Free 40S subunits and ribosomal half-mers (2.5 mer / disome) were quantified from polysome profiles as in panel (**b**) by calculating the ratio of the corresponding areas under the curve; shown are mean values ± SD ($n$ = 4 biological replicates; $p$-values determined by two-tailed, paired $t$-test). Source data are provided as a Source Data file.

During the initial phase of RQC, Hel2/ZNF598 recognizes collided ribosomes and marks them by RPS ubiquitination. Subsequently, the ribosome quality control trigger complex RQT (also termed ASC-1 complex in mammals) splits the 40S and 60S subunits of collided ribosomes, and half-mers form upon dissociation of the 60S subunit through the activity of RQT[14,16,20]. In principal, there are two possibilities how RNF10 could suppress half-mer formation: Either, RNF10 inhibits RQT complex-mediated dissociation of 60S from the leading stalled ribosome, or RNF10 promotes dissociation of the remaining 40S subunit. To test whether the first hypothesis might be true, we made use of a FACS-based RQC reporter system[37].

As expected[37], KD of ZNF598 specifically reduces RQC efficiency on the poly-lysine $(K^{AAA})_{20}$-containing reporter mRNA that imposes ribosome stalling (Supplementary Fig. 2e, f, visible as elevated mCherry / GFP ratio). In contrast, KD of RNF10 has no impact on RQC efficiency (Supplementary Fig. 2e, f), consistent with previous findings[26]. Since 60S dissociation is key to efficient RQC, this result indicates that RNF10 does not affect RQT complex-mediated 60S dissociation. This notion is supported by a recent publication showing directly by in vitro reconstitution of stalled ribosome complexes that RNF10 does not affect the activity of the mammalian RQT/ASC-1 complex[28]. Together, these data argue for the second hypothesis, whereby RNF10 promotes the dissociation of 40S subunits from half-mer intermediates during the resolution of stalled ribosome collision complexes.

Given that perturbation of protein synthesis elicits a multi-faceted stress response that interferes with cell cycle progression[38,39], we tested whether RNF10 KO would have an impact on cell proliferation. At a low dose of 0.02 μg/ml, ANI caused only a mild reduction in total protein synthesis by about 20% as measured by puromycin incorporation, while cell proliferation was strongly reduced with a > 80% reduction in histone H3 serine 10 phosphorylation as a marker of mitotic cells (Fig. 2f, Supplementary Fig. 2g). This result illustrates that cells respond to mild perturbation of protein synthesis by actively downregulating proliferation, reflecting activation of the ribotoxic stress response. We then cultured control and RNF10-KO cells over a period of 72 h in the absence or presence of low dose ANI, and cell numbers were recorded. Under basal conditions, the proliferation rate in RNF10-KO cells was reduced by ∼20% (Fig. 2g). In the presence of ANI, RNF10-KO cells showed a much stronger defect with a > 60% reduction in cell proliferation (Fig. 2g). In both cases, the proliferation rate was restored to the original level observed in parental cells by reintroduction of RNF10-WT into the KO cells. These results demonstrate that RNF10 plays an essential role in sustaining cell proliferation during chronic translation elongation stress.

## Ubiquitination of RPS3 at K214 promotes dissociation of stalled 40S subunits

In yeast, Mag2 was shown to monoubiquitinate RPS3 (uS3) at lysine (K) 212[21], and the corresponding position K214 in human RPS3 was found to be monoubiquitinated by RNF10 in HEK293 cells[26]. To test if RNF10 promotes dissociation of stalled 40S subunits through ubiquitination of human RPS3 at K214, we generated HeLa cells stably expressing FLAG-tagged RPS3-WT or RPS3-K214A by lentiviral transduction. As expected, FLAG-RPS3-WT was ubiquitinated upon ANI treatment, whereas FLAG-RPS3-K214A was not (Fig. 3a).

In these cells, we then suppressed expression of endogenous RPS3 by two consecutive transfections with an siRNA (S292) targeting the 5′ untranslated region, which does not affect expression of transgenic FLAG-RPS3-WT/K214A. Cells were then exposed to an intermediate (0.1 μg/ml for 2 h) or low dose (0.02 μg/ml for 48 h) of ANI before polysome profiles were recorded (Fig. 3b, controls in Supplementary Fig. 3). The first observation we made was that the peak of free 40S subunits was elevated in HeLa-FLAG-RPS3-K214A cells compared to HeLa-FLAG-RPS3-WT (Fig. 3b, quantified in the left panel of Fig. 3c), in line with the notion that RNF10 promotes turnover of stalled 40S subunits via RPS3 ubiquitination at K214[26].

We further noticed in this setup that HeLa-FLAG-RPS3-WT cells could more efficiently disassemble stalled ribosomes since half-mer formation was not observed upon ANI treatment (Fig. 3b, compare to parental HeLa in Fig. 2b and c). In contrast, half-mers were visible in the HeLa-FLAG-RPS3-K214A cells (Fig. 3b, quantified in the right panel of Fig. 3c). Elevated half-mer levels in HeLa-FLAG-RPS3-K214A cells provide evidence that dissociation of ribosomal half-mers is mediated by ubiquitination of RPS3 at K214.

## Perturbation of 40S ribosome biogenesis leads to proteasomal degradation of RNF10

Given its role in the dissociation and clearance of stalled 40S subunits, we wanted to explore whether RNF10 might play a role in the cellular response to the occurrence of defective ribosomes. To this end, we suppressed the synthesis of RPS3 by transfection of siRNA (#292) into HeLa cells for 16 h. As expected, this led to a decrease in the pool of free 40S subunits, a corresponding accumulation of free 60S subunits, and loss of actively translating ribosomes reflected by a decrease in mono- and polysomes (Fig. 4a). RPS3 protein levels were reduced in

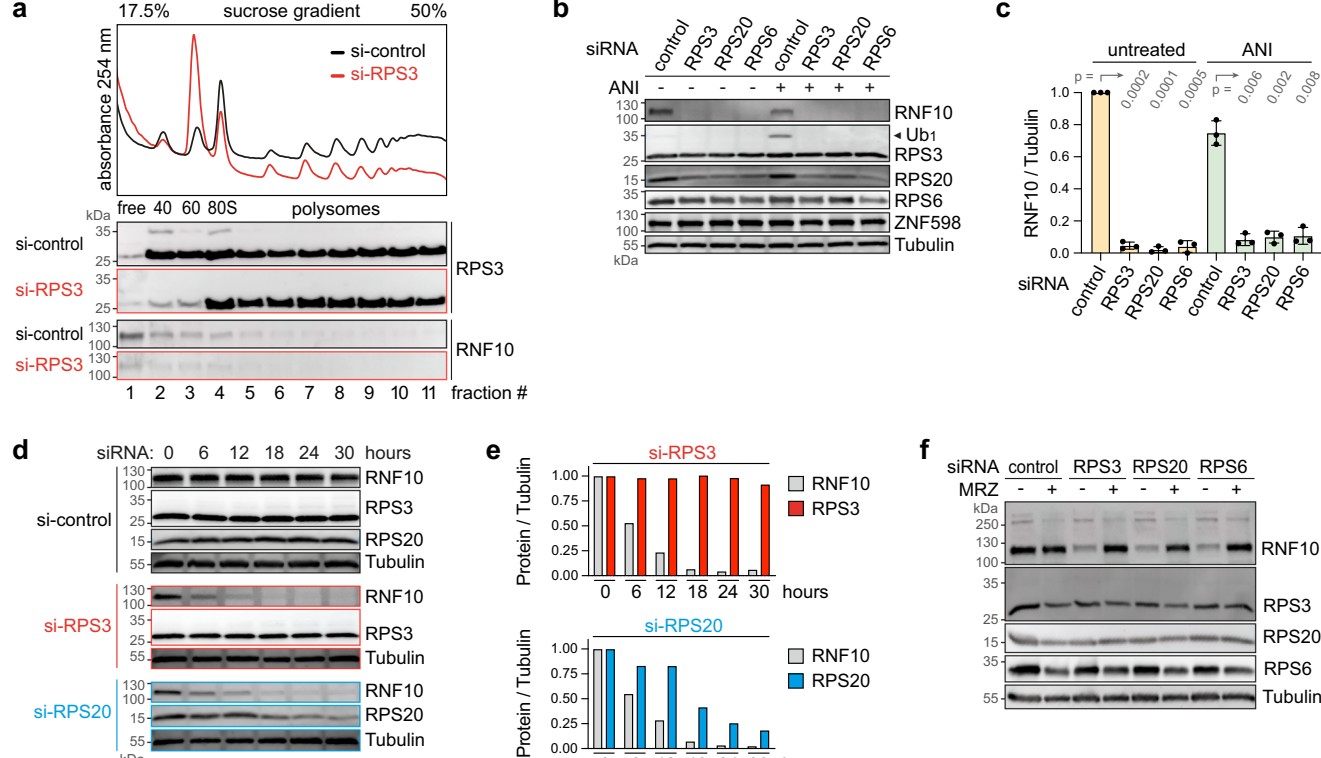

**Fig. 4 | Proteasomal degradation of RNF10 upon RPS KD. a** Polysome profile and fractionation analysis of HeLa cells 16 h after KD of RPS3 (siRNA S292) in comparison to control KD (S75); the distribution of RPS3 and RNF10 was assessed by Western blot analysis. The blot is representative of one experiment. **b** Western blot analysis of RNF10 and Ub$_1$-RPS3 in HeLa cells 24 h after control KD (siRNA S75) in comparison to KD of RPS3 (S292), RPS20 (S297) and RPS6 (S299), ± ANI treatment (2 h, 0.1 μg/ml). **c** RNF10 protein levels were quantified from Western blots as in panel (**b**), shown are mean values ± SD (*n* = 3 biological replicates; *p*-values determined by two-tailed, paired *t*-test). **d** Western blot analysis for RNF10, RPS3 and

RPS20 was carried out 0, 6, 12, 18, 24 and 30 h after control KD (S75), KD of RPS3 (S292) or KD of RPS20 (S297) in HeLa cells. **e** RNF10, RPS3 and RPS20 protein levels were quantified from the Western blot in panel (**d**) (*n* = 1 biological replicate). **f** Western blot analysis of RNF10 6 h after control KD (S75) or KD of RPS3 (S292), RPS20 (S297) or RPS6 (S299), ± simultaneous treatment with the proteasome inhibitor marizomib (MRZ, 6 h, 100 nM). The blot is representative of one experiment. In panels (**b**–**f**), Tubulin serves as loading control. Source data are provided as a Source Data file.

polysome profile fractions #2 and #3 corresponding to the free 40S pool, while the reduction was barely visible in the Western blot of total protein lysates (Fig. 4b, Supplementary Fig. 4a, b), and not visible in polysome profile fractions #4–11 (Fig. 4a). This indicates that RPS3 is very stable as part of intact 40S subunits that participate in active translation (80S and polysomes), while newly assembled small ribosomal subunits lacking RPS3 are restricted to the free 40S fraction and are not translation-competent. Efficient KD of RPS3 mRNA was also confirmed by quantitative PCR (Supplementary Fig. 4c).

To our surprise, RNF10 protein levels were abolished by KD of RPS3 (Fig. 4b, quantified in Fig. 4c). This effect occurred even when we transfected RPS3 siRNA (#292) at concentrations as low as 10 nM (Supplementary Fig. 4d). Likewise, RNF10 was no longer detectable when two different siRNAs (#293 and #294) against RPS3 were used (Supplementary Fig. 4a and e).

To pursue this observation, we explored whether the effect is specific for RPS3 KD, or whether 40S biogenesis perturbation in general causes a drop in RNF10 levels. Indeed, KD of RPS6, RPS19 and RPS20 led to the same drop in RNF10 levels as KD of RPS3 (Fig. 4b and c, and Supplementary Fig. 4f), indicating that this is a general response to 40S biogenesis perturbation (Supplementary Fig. 4g, h). Loss of RNF10 was also observed at the functional level since ubiquitination of RPS3 upon ANI treatment was abolished upon KD of RPS3, RPS6 and RPS20 (Fig. 4b, and Supplementary Fig. 4a, b). Interestingly, the levels of the E3 ligase ZNF598 remained unchanged after KD of RPS3, RPS6 or RPS20 (Fig. 4b).

To determine how rapidly RNF10 is degraded after inhibition of 40S protein synthesis, we performed time course experiments upon KD of RPS3 and RPS20 (Fig. 4d). While RNF10 disappears within ~18 h, total RPS20 levels decline to about 50% after 18 h of KD, and the decay of total RPS3 is barely visible over the 30 h time course (Fig. 4e). This result indicates that RNF10 degradation is linked to the appearance of defective 40S subunits rather than to the decline of total RPS3/20 protein levels.

Finally, we explored the mechanism by which RNF10 is downregulated upon 40S biogenesis perturbation. qPCR analysis showed no major change in RNF10 mRNA levels upon KD of RPS3 (Supplementary Fig. 4i), suggesting that RNF10 is not subject to transcriptional control under these conditions. We then tested if RNF10 might be degraded via the proteasome, and treated HeLa cells with the proteasome inhibitor marizomib (MRZ; also termed Salinosporamide A, NPI-0052), which irreversibly inhibits all three proteasomal activities[40]. Indeed, MRZ treatment prevented the loss of RNF10 protein upon KD of RPS3, RPS6 and RPS20 (Fig. 4f), demonstrating that 40S biogenesis perturbation causes proteasomal degradation of RNF10.

## Perturbation of 60S ribosome biogenesis causes accumulation of RNF10

Prompted by the above observation, we next asked whether perturbation of 60S biogenesis would also affect RNF10 levels and/or activity. To this end, a protein of the large ribosomal subunit, RPL7, was knocked down, causing a decrease in free 60S subunits, an

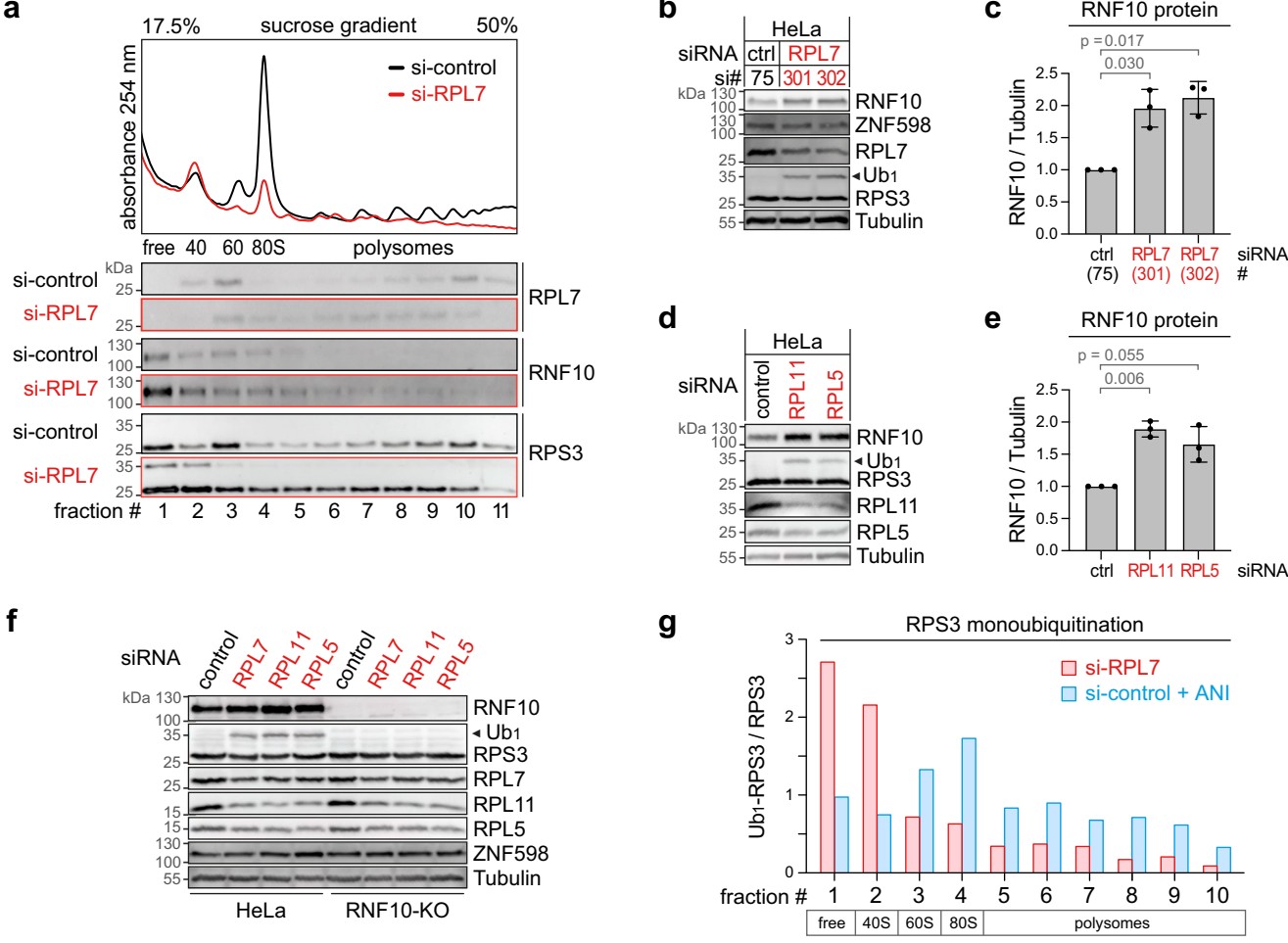

**Fig. 5 | Accumulation of RNF10 upon RPL KD. a** Polysome profile and fractionation analysis of HeLa cells 16 h after KD of RPL7 (siRNA S301) in comparison to control KD (S75); the distribution of RPS3 and RNF10 was assessed by Western blot analysis. The blot is representative of 3 independent experiments. **b** Western blot analysis of RNF10 and Ub$_1$-RPS3 in HeLa cells 24 h after KD of RPL7 using two different siRNAs in comparison to control KD. **c** RNF10 protein levels were quantified from Western blots as in panel (**b**), shown are mean values ± SD ($n = 3$ biological replicates; $p$-values determined by two-tailed, paired $t$-test). **d** Western blot analysis of RNF10 and Ub$_1$-RPS3 in HeLa cells 24 h after KD of RPL11 (S304) and RPL5 (S310) in comparison to control KD (S75). **e** RNF10 protein levels were quantified from Western blots as in panel (**d**), shown are mean values ± SD ($n = 3$ biological replicates; $p$-values determined by two-tailed, paired $t$-test). **f** Western blot analysis of RNF10 and Ub$_1$-RPS3 in HeLa and HeLa-RNF10-KO cells 24 h after KD of RPL7 (S301), RPL11 (S304) or RPL5 (S310) in comparison to control KD (S75). The blot is representative of one experiment. **g** The ratio of Ub$_1$-RPS3 / RPS3 protein levels ($n = 1$ biological replicate) in parental HeLa cells 24 h after KD of RPL7 (S301) or control KD (S75) was calculated by quantification from Western blots in (Supplementary Fig. 5d). In panels (**b**–**f**), Tubulin serves as loading control. Source data are provided as a Source Data file.

accumulation of free 40S subunits, and again a loss of actively translating ribosomes indicated by the reduction in mono- and polysomes (Fig. 5a). Interestingly, this had the opposite effect as RNF10 protein levels were elevated by ~2-fold after knocking down RPL7 – an effect we observed with two different siRNAs (#301 and #302, Fig. 5b, quantification in Fig. 5c). Accordingly, RNF10 activity was elevated under these conditions, as reflected by an increase in Ub$_1$-RPS3 (Fig. 5a, b). KD of RPL7 did not affect RNF10 mRNA levels (Supplementary Fig. 5a), similar to what we observed upon KD of RPS3 (Supplementary Fig. 4i), indicating that RNF10 is not regulated at the level of transcription or mRNA stability. Of note, the levels of ZNF598 were not affected by RPL7 KD (Fig. 5b, Supplementary Fig. 5b).

We then tested other proteins of the large ribosomal subunit, and found that KD of RPL5 and RPL11 led to a similar increase of RNF10 levels by ~1.6- to 1.9-fold (Fig. 5d, quantification in Fig. 5e), as well as enhanced monoubiquitination of RPS3 (Fig. 5d). In HeLa RNF10-KO cells, Ub$_1$-RPS3 was not induced by RPL7, RPL11 or RPL5 KD (Fig. 5f), demonstrating that RNF10 is responsible for RPS3 monoubiquitination also under these conditions. As seen with the KD of RPL7, KD of RPL5

and RPL11 led to a pronounced reduction in polysomes, a reduction of the free 60S pool and an increase of 40S subunits, indicative of impaired 60S ribosome biogenesis (Supplementary Fig. 5c).

By performing Western blot analysis after polysome fractionation, we observed that RPS3 monoubiquitination upon RPL7 KD is primarily visible in the first two fractions associated with free protein and 40S subunits, whereas ANI-induced monoubiquitination of RPS3 has its maximum in the 80S fraction and extends far into polysomes (Fig. 5g, Supplementary Fig. 5d). This is compatible with the idea that RNF10 targets 40S subunits that are stalled at the level of translation initiation when 60S biogenesis is impaired upon RPL7 KD, whereas RNF10 targets 40S subunits of ribosomes stalled during elongation when cells are exposed to ANI. Accumulation of the ribosome collision sensor EDF1[35] in polysomal fractions upon ANI treatment, but not RPL7 KD (Supplementary Fig. 5d), confirms this notion.

One possible explanation for the elevated RNF10 levels is that the protein is stabilized upon RPL KD. To address this possibility, we performed KDs of RPL7, RPL11 and RPL5 in the absence and presence of the proteasome inhibitor MRZ (Supplementary Fig. 5e). Curiously,

MRZ treatment led to a slight decrease of RNF10 levels (Supplementary Fig. 5f, g) although total K48-linked polyubiquitinated proteins showed strong accumulation as a read-out for successful proteasome inhibition (Supplementary Fig. 5f). While the differences between −MRZ and +MRZ were statistically not significant, the pattern indicates that proteasome inhibition may interfere with the induction of RNF10 protein levels upon RPL KD. This is in line with the observation that MRZ treatment prevented RPS3 monoubiquitination upon RPL KD (Supplementary Fig. 5f), suggesting that proteasome inhibition interferes with the recognition and/or ubiquitination of 40S subunits stalled at the level of initiation. Since there is no increase in RNF10 levels upon MRZ treatment under control conditions (Supplementary Fig. 5f, g), we concluded that RNF10 is normally not subjected to proteasomal decay. Hence it is unlikely that the increase in RNF10 levels upon RPL KD is caused by further stabilization.

### Transcription factor p53 does not control RNF10 expression

Ribosome biogenesis defects including mutations in ribosomal protein genes are known to cause activation of the transcription factor p53, mainly through inhibition of the p53-specific E3 ubiquitin ligase MDM2 by elevated levels of extra-ribosomal RPS and RPL proteins[41,42]. Given the connection between ribosome biogenesis stress and p53, we examined whether changes in RNF10 expression are controlled by p53. To this end, we knocked down RPS3, RPS6 and RPS20 as well as RPL7 in human HCT116 colorectal cancer cells that are either WT or deficient for p53[43]. As in HeLa cells, RNF10 levels in HCT116 cells were decreased upon KD of RPS3, RPS6 and RPS20, and increased upon RPL7 KD (Fig. 6a, quantification in Fig. 6b), although to a lesser degree than in HeLa cells (Supplementary Fig. 6a, b). We did not observe differences in the change of RNF10 levels between the HCT116 p53[+/+] and p53[-/-] cells (Fig. 6a, b), showing that RNF10 expression is regulated independently of p53. Thus, the RNF10 response to ribosome biogenesis perturbation does not appear to be linked to p53 signaling.

### RNF10 promotes 40S dissociation upon 60S ribosome biogenesis perturbation

Similar to treatment with ANI or CHX at low to intermediate doses (Fig. 2b–e and Supplementary Fig. 2b–d), KD of RPL7 or RPL11 led to pronounced ribosomal half-mer formation (Figs. 5a and 7a). In the case of ANI or CHX, and in analogy to yeast[14], this is likely due to 60S dissociation upon elongation stalling mediated by the RQT/ASC-1 complex[16,20]. In the case of RPL KD, half-mer formation results from depletion of the 60S pool and the ensuing stoichiometric imbalance between 60S and 40S subunits, whereby 40S ribosomal subunits are stalled at the start codon "waiting" for 60S subunit joining. The notion of elongation versus initiation stalling is supported by our observation that RPS3 monoubiquitination extends far into the polysomal part of the gradient upon elongation stalling when cells are treated with ANI, whereas $Ub_1$-RPS3 is restricted to the first few fractions of polysome gradients upon initiation stalling when cells are depleted of 60S subunits (Fig. 5g, Supplementary Fig. 5d).

In HeLa cells lacking RNF10, the distribution of ribosomal subunits upon RPL7 or RPL11 KD was not altered in comparison to parental HeLa cells (Supplementary Fig. 7a, quantification in Supplementary Fig. 7b). In cells with enhanced expression of RNF10 (KO+RNF10-WT), however, ribosomal half-mer formation upon RPL7 or RPL11 KD was strongly reduced, while it remained unaffected in KO+RNF10-Mut cells lacking RNF10 E3 ubiquitin ligase activity (Fig. 7a, b). Interestingly, elevated expression of RNF10 in the KO+RNF10-WT cells increases the 60S/40S ratio (Supplementary Fig. 2a, visible also in Fig. 7a and c) and thereby counteracts the 60S/40S imbalance induced by RPL7 KD, leading to a delayed formation of half-mers upon RPL7 KD (Fig. 7c). These experiments reveal an important role of RNF10 in the maintenance of ribosomal subunit stoichiometry by promoting the dissociation and clearance of 40S subunits that are stalled during ribosomal initiation or elongation.

## Discussion

Various quality control pathways monitor the process of protein biosynthesis and thereby ensure maintenance of cellular homeostasis[1,2]. Insults that interfere with translation elongation such as exposure to ribotoxins, defective ribosomes, aberrant mRNAs or tRNA malfunction all lead to ribosome stalling, triggering a cascade of events that promote dissociation of the stalled ribosomal subunits and degradation of the nascent polypeptide chain via RQC[6–8,11], degradation of the affected mRNA via no-go or non-stop mRNA decay[17,18] and/or degradation of non-functional rRNAs via NRD[15]. Ubiquitination of small ribosomal proteins is central to the initiation and proper execution of all these interconnected quality control pathways[6,15].

Here, we show that human RNF10 is responsible for mono-ubiquitinating RPS3 upon treatment with different translation elongation inhibitors in HeLa and non-cancerous RPE1 cells (Fig. 1, Supplementary Fig. 1), consistent with previous reports on ubiquitination of RPS3 and RPS2 (uS5) by RNF10 in HEK293/T cells under conditions of initiation and elongation stalling[26,27]. RNF10 specifically targets K214 within RPS3[26] (Fig. 3), which corresponds to K212 of yeast RPS3 thus demonstrating that mammalian RNF10 is the functional homologue of yeast Mag2[21]. While RPS3 has various extraribosomal functions[31], we could show that RNF10 monoubiquitinates mainly 40S- and ribosome-associated RPS3 upon ANI treatment (Fig. 1c, d). Interestingly, we did not observe a change in the distribution of RNF10 upon ANI treatment, with RNF10 being present in both the free and ribosome-associated fractions (Fig. 1c), indicating that the interaction of RNF10 with ribosomes might be rather transient.

Beyond revealing the loss of RPS3 monoubiquitination as a molecular phenotype[26,27], our study demonstrates that RNF10-KO cells have a functional defect in the resolution of stalled 40S subunits, and a corresponding proliferation defect that is strongly aggravated by chronic translation elongation stress (Fig. 2).

Several studies have shown that the collision interface of the stalled and the collided ribosome is recognized by RQC factors, which induce ribosome ubiquitination and disassembly[12–14,16,19,20]. While treatment of cells with a high dose of ANI or CHX does not lead to ribosomal half-mer formation since wide-spread stalling of elongating ribosomes prevents collisions, we observed efficient formation of ribosomal half-mers after provoking ribosome collisions through treatment with the translation elongation inhibitors at low to intermediate doses (Fig. 2 and Supplementary Fig. 2). Half-mer formation under these conditions can be explained by a recently proposed mechanism for the clearance of the stalled 80S-80S di-ribosome collision complex. Herein, the di-ribosome is processed into an 80S-40S half-mer intermediate by the RQT/ASC-1 complex before it is fully cleared[14].

Our results demonstrate that RNF10 KO causes elevated half-mer formation in ANI-treated cells (Fig. 2c and e), providing evidence that RNF10 plays an essential and non-redundant function in the resolution of ribosomal collisions when translation elongation is perturbed (Fig. 8a). Since RNF10 KO and the RPS3-K214A mutation both lead to similar phenotypes with elevated half-mer levels (Fig. 2 and Fig. 3), RPS3 monoubiquitination is a key event by which RNF10 promotes the dissociation of stalled 40 S subunits. This situation is mirrored by the effect of enhanced RNF10-WT expression, which represses polysomal half-mer formation upon elongation inhibition (Fig. 2b–e). Moreover, enhanced expression of RNF10 leads to a substantial decrease of free 40S subunits, and an increase of the free 60S pool (Supplementary Fig. 2a), as noted previously[26,27]. Together, these results indicate that ubiquitination of ribosomal proteins by RNF10 mediates the recognition, dissociation, and clearance of stalled ribosomal 40S subunits. As a consequence, ubiquitinated 40S subunits are either degraded by an

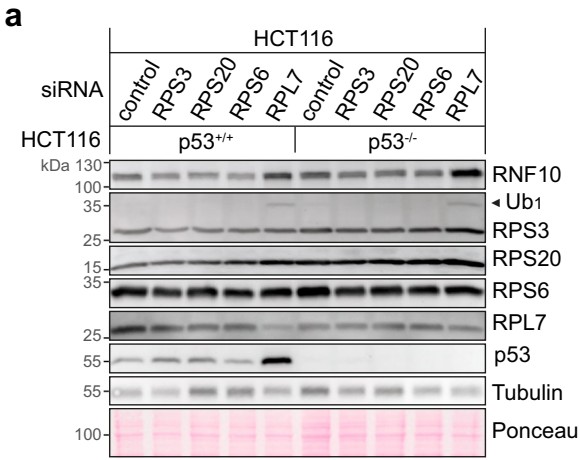

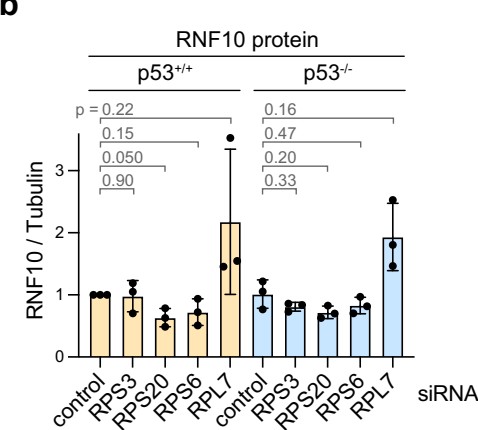

**Fig. 6 | Transcription factor p53 does not control RNF10 expression. a** Western blot analysis of RNF10 and Ub$_1$-RPS3 in HCT116 p53$^{+/+}$ and p53$^{-/-}$ cells 24 h after KD of RPS3 (siRNA S292), RPS20 (S297), RPS6 (S299) and RPL7 (S301) in comparison to control KD (S75); Tubulin and Ponceau staining serve as loading controls. **b** RNF10 protein levels were quantified from Western blots as in panel (**a**), shown are mean values ± SD ($n$ = 3 biological replicates; $p$-values determined by two-tailed, paired $t$-test). Source data are provided as a Source Data file.

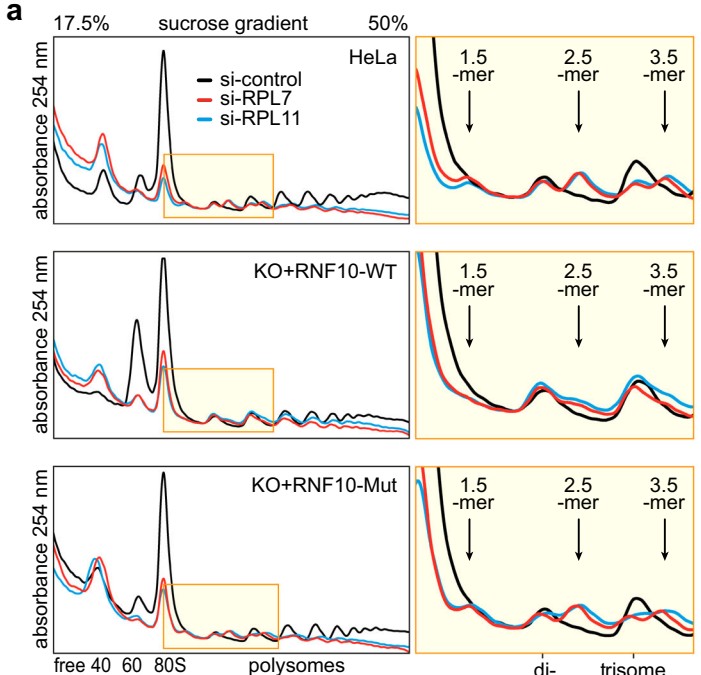

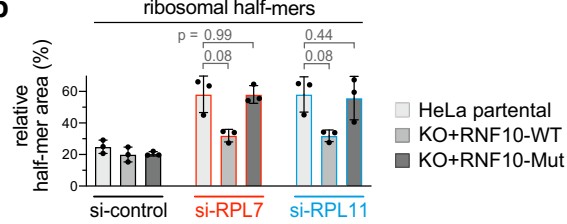

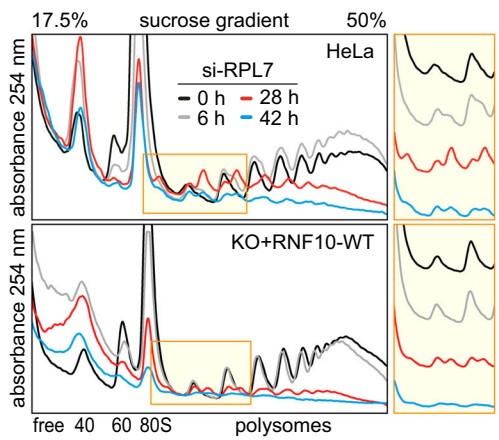

**Fig. 7 | RNF10 promotes 40S dissociation upon RPL KD. a** Polysome profile analysis of HeLa, HeLa-KO+RNF10-WT and HeLa-KO+RNF10-Mut (C225S/C228S) cells 16 h after KD of RPL7 (siRNA S301) or RPL11 (S304) in comparison to control KD (S75); the yellow area is enlarged on the right side. **b** Ribosomal half-mers (2.5-mer/disome ratio) were quantified from polysome profiles as in panel (**a**), shown are mean values ± SD ($n$ = 3 biological replicates; $p$-values determined by two-tailed, paired $t$-test). **c** Polysome profile analysis in parental HeLa cells and HeLa-KO +RNF10-WT cells following KD of RPL7 (S301) for different periods of time; profiles are shown separately in the yellow area on the right side. Source data are provided as a Source Data file.

autophagy-independent lysosomal pathway or rescued by the deubiquitinase USP10 as part of a recycling mechanism[26,27].

Our study further revealed that the enhanced expression of RNF10-WT also prevents polysomal half-mer formation upon perturbation of 60S ribosome biogenesis (Fig. 7a, b). Thus, RNF10 has the capacity to also promote dissociation of "idle" 40S (48S) subunits stalled at the level of translation initiation (Fig. 8b). This finding is in line with increased RPS3 ubiquitination described upon treatment with pharmacological inhibitors of translation initiation[27]. Since 40S stalling during initiation occurs in a spatial context that is very different from elongation stalling (Fig. 8a, b), it is currently unclear how RNF10 recognizes these structurally distinct[20,44] 40S subunits. One possibility is that RNF10 recognizes these 40S subunits through common features: under both conditions, the stalled 40S subunits are engaged with mRNA and their inter-subunit surface is not occupied by a 60S subunit. Since RPS3 is located in the mRNA entry channel of the 40S subunit and contacts the mRNA[20,45–47], it is well positioned to fulfil a monitoring function for the presence of mRNA. Moreover, the C-terminal tail of RPS3, which includes K214, extends into the cytoplasm[20,46], where it is accessible for ubiquitination by RNF10.

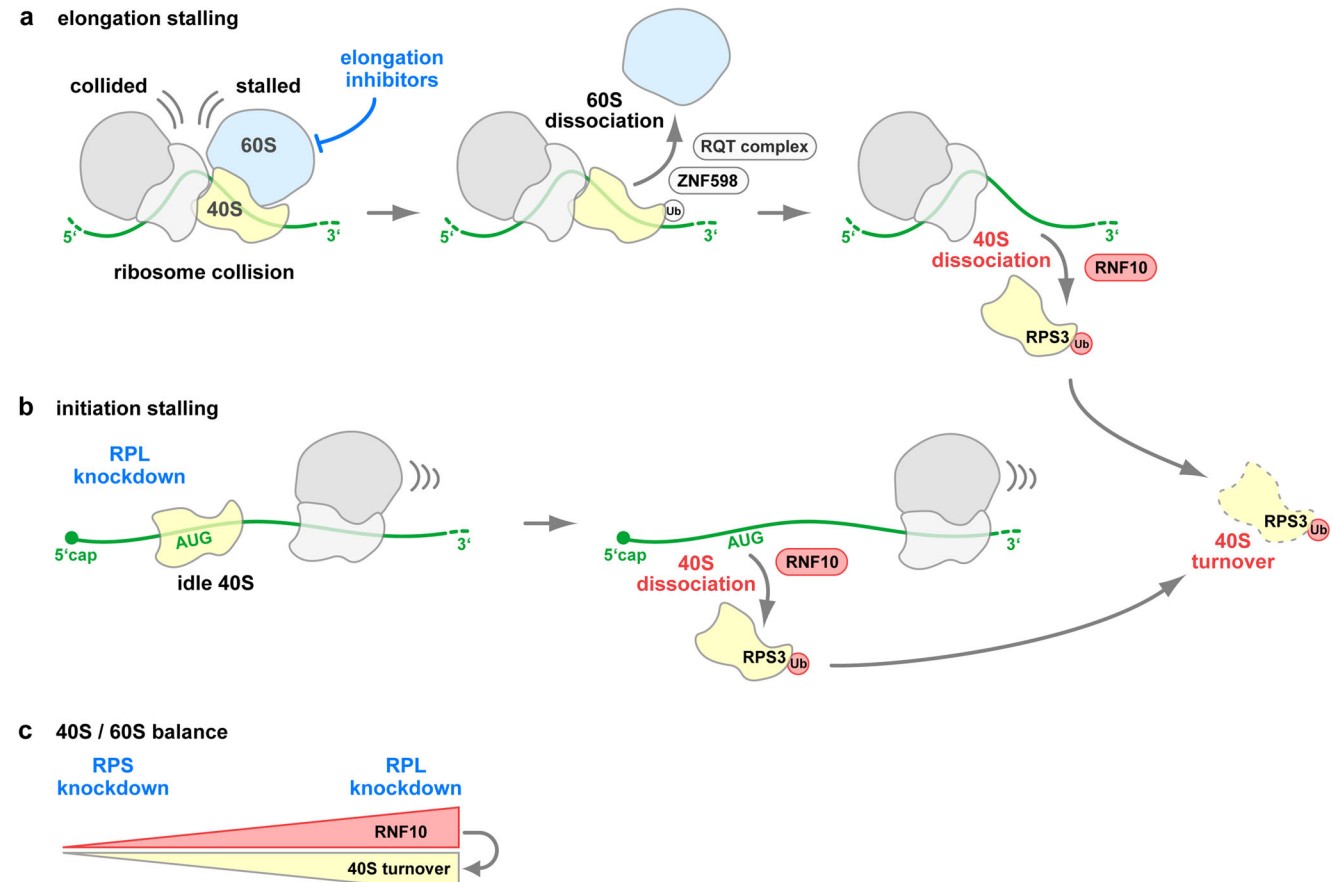

**Fig. 8 | Model of RNF10 promoting half-mer dissociation of stalled 40S subunits during translation elongation or initiation. a** Translation elongation inhibition leads to stalling and collision of elongating ribosomes. As a consequence, small ribosomal proteins are ubiquitinated by ZNF598, which leads to dissociation of the stalled 60S subunit and to a 80S-40S half-mer intermediate that remains associated with the mRNA. Monoubiquitination of ribosomal RPS3 by RNF10 then leads to dissociation of the 40S subunit from the half-mer intermediate. **b** Knockdown of large ribosomal proteins (RPL) leads to initiating 40S subunits waiting for 60S subunit joining at the start codon (AUG). These stalled initiating 40S subunits are monoubiquitinated on RPS3 by RNF10, promoting their dissociation from the mRNA. Following dissociation, 40S subunits monoubiquitinated on RPS3 are subjected to turnover. **c** RNF10 protein levels are downregulated after RPS KD and upregulated after RPL KD. This regulatory mechanism helps to maintain the stoichiometric balance between 40S and 60S subunits through RNF10-induced 40S turnover.

Interestingly, the C-terminal tail of RPS3 is in direct contact to RACK1 (Asc1 in yeast), a 40S protein required for the initiation of RQC[13,20,46]. Structural studies will be required to determine how distinct stalled 40S subunits can be recognized by RNF10.

Ribosome biogenesis occurs along a precise and highly orchestrated pathway, whose perturbation often results in the accumulation of specific rRNA precursors or the degradation of defective pre-40S or pre-60S particles[48]. Interestingly, impairment of 40S biogenesis via KD of small ribosomal proteins led to rapid degradation of RNF10 by the proteasome and failure to ubiquitinate RPS3 upon translation elongation inhibition (Fig. 4). Vice versa, impairment of 60S biogenesis by knockdown of large ribosomal proteins caused an elevation of RNF10 protein levels and monoubiquitination of RPS3 (Fig. 5), reflecting the excess of 40S subunits that are either free or engaged in nonproductive initiation events. By controlling the levels of RNF10, cells appear to have evolved a mechanism that helps maintain the stoichiometric balance between 40S and 60S subunits (Fig. 8c). Regulation of RNF10 protein levels upon ribosome biogenesis perturbation neither depends on the p53-dependent ribosome biogenesis stress response (Fig. 6), nor does it involve changes in RNF10 mRNA levels (Supplementary Fig. 4i, Supplementary Fig. 5a), indicating that transcription or mRNA stability are not involved. Further work will be needed to determine the molecular mechanism by which RNF10 levels are regulated under these conditions.

In conclusion, we could establish a tight connection between de novo synthesis of ribosomal subunits and RNF10, whereby the size of the free 40S subunit pool correlates with RNF10 levels. One possible explanation could be that RNF10 associates with, and is stabilized by, newly synthesized 40S subunits. RNF10 might take part in monitoring the functionality of such juvenile 40S subunits in a translation-coupled process, whereby activation of RNF10 upon 40S stalling promotes the dissociation and clearance of defective subunits. Our findings demonstrate that RNF10 functions at the nexus of ribosome biogenesis and surveillance of ribosome functionality, pointing towards a key role in elongation RQC, in the quality control of initiating 40S subunits and in monitoring ribosomal subunit stoichiometry.

## Methods

### Cell culture

HeLa, HEK293 and HCT116 cells were cultured in Dulbecco's Modified Eagle Medium (DMEM; Gibco), RPE1 cells in DMEM/Ham's F-12 Nutrient Mix (1:1; Gibco). Both media were supplemented with 10% [v/v] fetal bovine serum (Sigma), 2 mM L-glutamine, 100 U/ml penicillin and 100 μg/ml streptomycin (all PAN Biotech). The cells were cultured at subconfluency at 37 °C in a 5% $CO_2$ incubator. The PCR Mycoplasma Test Kit (AppliChem) was used to regularly test all cell lines for mycoplasma contamination.

## siRNA transfection

KD experiments were carried out via reverse transfection using 100 nM siRNA according to the manufacturer´s instructions with Lipofectamine RNAiMAX transfection reagent (Thermo Fisher Scientific) for 96 h. For KD of RPS and RPL proteins, siRNAs were transfected for 24 h if not indicated differently. The non-targeting control siRNA S75 was purchased from Qiagen (10278210), all other siRNAs from Eurofins MWG Operon. siRNAs were designed with the support of the siDESIGN Center (Horizon Discovery). siRNA sequences:

S193, 5′-GCACCAAGTCCAAGAAGAAC-3′ (ZNF598)
S201, 5′-CCUCGACAAAUGGUCCUGU-3′(ZNF598)
S213, 5′-GGAUUGAGAUAGAGGAGAA-3′ (RNF10)
S214, 5′-GAGAAGACGUGGAGUAAAU-3′ (RNF10)
S292, 5′-GCGGCAAGAUGGCAGUGCA-3′ (RPS3)
S293, 5′-CAGAAGAGGUUUGGCUUUC-3′ (RPS3)
S294, 5′-GCGGAGACCCUGUUAACUA-3′ (RPS3)
S297, 5′-GCACAGUCCUUCUGAGA-3′ (RPS20)
S299, 5′-GUGAAGAAUGGAAGGGUUA-3′(RPS6)
S301, 5′-CGAAUUGCUUUGACAGAUA-3′ (RPL7)
S302, 5′-CGUCAAAUCUUCAAUGGAA-3′ (RPL7)
S304, 5′-GGGCCAAGGCAGAAGAAAU-3′ (RPL11)
S309, 5′-GGCUGAAAAUGGUGGAAAA-3′ (RPS19)
S310, 5′-AGGCCUACUUUAAGAGAUA-3′ (RPL5)

## Plasmid transfection

HeLa and HEK293 cells were seeded at a density of $1.5 \times 10^6$ cells/dish in 10 cm dishes. On the following day, cells were transfected with plasmid DNA diluted in Opti-MEM Reduced Serum Medium (Thermo Fisher Scientific) using Lipofectamine 3000 transfection reagent (Thermo Fisher Scientific) at a ratio of 1:3. The medium was exchanged 4 h post transfection.

## Plasmid generation

The plasmids pcDNA3-HA (p2003)[49] and pcDNA3-FLAG (p2530)[50] have been described previously. pWPI-BLR (p3784), pCMV-Δ8.91 (p3785) and pMD2.G (p3786) were generously provided by Alessia Ruggieri (Center for Integrative Infectious Disease Research, University Hospital Heidelberg). Plasmid pcDNA3-RPS3-FLAG-TEV-SBP (p3437) was generated by Bogdan Jovanovic (MI3, Heidelberg University). Plasmids pmGFP-P2A-3xFLAG-K(AAA)$_0$-P2A-mCherry (p3878) and pmGFP-P2A-3xFLAG-K(AAA)$_{20}$-P2A-mCherry (p3880)[37] were kindly provided by Ramanujan Hegde (MRC Laboratory of Molecular Biology, Cambridge, UK).

To generate pcDNA3-HA-RNF10-WT (p3892), the RNF10 cDNA was PCR amplified with the primers G5722/G5723 from pENTR221-RNF10 (p3905; purchased from BioCat) and inserted into pcDNA3 (p2003) via KpnI/EcoRI. RNF10 cDNA carrying the C225S/C228S mutations was then amplified with primers G5724/G5725 from p3892 to generate pcDNA3-HA-RNF10-Mut (p3894). For pWPI-BLR-HA-RNF10-WT (p3898) and pWPI-BLR-HA-RNF10-Mut (p3900), the HA-tagged RNF10 cDNA was amplified from p3892 and p3894, respectively, using Gibson overhang primers G5712/G5728, and inserted into pWPI-BLR (p3784) via SmaI through Gibson assembly.

To generate pcDNA3-FLAG-RPS3-WT (p3890), the RPS3 cDNA was PCR amplified from plasmid pcDNA3-RPS3-FLAG-TEV-SBP (p3437) using primers G5716/G5717, and inserted into pcDNA3 (p2530) via BamHI/EcoRI. RPS3 cDNA carrying the K214A mutation was then amplified with primers G5718/G5719 from p3890 to generate pcDNA3-FLAG-RPS3-K214A (p3891). For pWPI-BLR-FLAG-RPS3-WT (p3896) and pWPI-BLR-FLAG-RPS3-K214A (p3897), the FLAG-tagged RPS3-WT and -K214A cDNA was PCR amplified from p3890 and p3891, respectively, using Gibson overhang primers G5712/G5728, and inserted into pWPI-BLR via SmaI through Gibson assembly.

For pSpCas9(BB)-2A-GFP-*RNF10*-guide29 (p3906) and pSpCas9(BB)-2A-GFP-*RNF10*-guide81 (p3907), oligonucleotides corresponding to the guide RNA (gRNA) targeting RNF10 locus 29 (G5193, G5194) and 81 (G5195, G5196) were first 5′ phosphorylated with T4 polynucleotide kinase (NEB). The 5′-phosphorylated sense and antisense oligonucleotides were then purified by phenol-chloroform extraction, annealed and inserted into the BpiI site of plasmid pSpCas9(BB)-2A-GFP (PX458)[51] (p3511) to generate pSpCas9(BB)-2A-GFP-*RNF10*-guide29 (p3906) and pSpCas9(BB)-2A-GFP-*RNF10*-guide81 (p3907).

## CRISPR/Cas9-mediated genome editing

Guide RNA sequences for targeting the RNF10 locus and primers for the validation PCR were selected with help of the E-CRISP design tool (German Cancer Research Center; http://www.e-crisp.org/E-CRISP/)[52].

G5193, 5′-CACCGTAGGTGGATAGAGGCATAT-3′ (gRNA locus 29 rev)
G5194, 5′-AAACATATGCCTCTATCCACCTAC-3′ (gRNA locus 29 fwd)
G5195, 5′-CACCGTGGAAGACGAGATGAGGTA-3′ (gRNA locus 81 fwd)
G5196, 5′-AAACTACCTCATCTCGTCTTCCAC-3′ (gRNA locus 81 rev)
G5201, 5′-ATGGTTTCCTTGATGCTTTGCT-3′ (PCR primer locus 29 fwd)
G5202, 5′-TGTACCCAGGATTATTTGTCCCC-3′ (PCR primer locus 29 rev)
G5426, 5′-CTCTTCTGTCCTGTTACCAGG-3′ (PCR primer locus 81 fwd)
G5427, 5′-CTCTCAAAATCCACCTCCTC-3′ (PCR primer locus 81 rev)

For the generation of RNF10-KO cells, HeLa cells were transfected with the Cas9-gRNA expressing plasmid pSpCas9(BB)-2A-GFP-*RNF10*-guide29 (p3906) or pSpCas9(BB)-2A-GFP-*RNF10*-guide81 (p3907) using Lipofectamine 3000 transfection reagent (Thermo Fisher Scientific) at a ratio of 4:1. GFP-positive single cells were sorted into 96-well plates containing 10% [v/v] conditioned medium by FACS (FlowCore, Medical Faculty Mannheim) 3 days after transfection. Loss of RNF10 expression in single cell clones was tested by Western blot analysis and sequencing of the genomic locus. The HeLa RNF10-KO clones guide29-clone7 (29-7) and guide81-clone7 (81-7) were selected for single clone analysis; clone 29-7 was also used to generate the rescue cell lines HeLa-KO+RNF10-WT and HeLa-KO+RNF10-Mut. For the generation of RNF10-KO pools, guide29-clone 3, 4, 6, 7 and 12 were combined for the 29-pool, and guide81-clone 4, 7, 10, 12 and 15 were combined for the 81-pool. The HeLa KO-control clone underwent the same CRISPR/Cas9 procedure but without gRNA.

## Lentiviral transduction

For stable expression of HA-RNF10-WT, HA-RNF10-Mut (C225S/C228S), FLAG-RPS3-WT and FLAG-RPS3-K214A, lentiviral particles were produced in HEK293 cells via co-transfection of the lentiviral expression plasmids pWPI-HA-RNF10-WT (p3898), pWPI-HA-RNF10-Mut (C225S/C228S, p3900), pWPI-FLAG-RPS3-WT (p3896) or pWPI-FLAG-RPS3-K214A (p3897) together with the lentiviral packaging vector pCMV-Δ8.91 (p3785) and the VSV-G expression vector pMD2.G (p3786) at a ratio of 3:3:1 using TurboFectin 8.0 (OriGene). The culture medium was exchanged 6 h post transfection, and after 2 days, supernatants enriched with lentiviral particles were transferred onto the target HeLa cells after passage through a 0.45 μm syringe filter. A second transfer was repeated 12 h later. Transduced cells were washed with culture medium 2 days after the second transduction and selected with 10 μg/ml blasticidine S (AppliChem) for 1 week. HeLa-RNF10-KO (29-7) cells were used for expressing HA-RNF10-WT and HA-RNF10-Mut, generating HeLa-KO+RNF10-WT and Hela-KO+RNF10-Mut. Parental HeLa cells were used for expressing FLAG-RPS3-WT and FLAG-RPS3-K214A, generating HeLa-RPS3-WT and HeLa-RPS3-K214A.

## Western blot analysis

Cells were trypsinized, resuspended with culture medium, washed with phosphate-buffered saline (PBS) and lysed in ice-cold RIPA protein lysis buffer (50 mM Tris-HCl pH 7.4, 150 mM NaCl, 15 mM MgCl$_2$, and 1% Triton X-100) supplemented with EDTA-free protease inhibitor

cocktail (Complete Mini, Roche) and phosphatase inhibitor cocktail (Phosstop Easypack, Roche). Samples were tumbled for 10 min at 4 °C and sonicated (5 cycles, each 30 s on/30 s off, high strength) in a Bioruptor (Diagenode). 10–20 µg total protein were mixed with 5× or 2× SDS Laemmli buffer, loaded onto 12% polyacrylamide gels, separated in Tris-glycine SDS running buffer (25 mM Tris, 250 mM glycine, 0.1% [w/v] SDS) at 30 mA and transferred onto nitrocellulose membrane (0.2 µm pore size, PeqLab) in Tris-glycine blotting buffer (20 mM Tris, 150 mM glycine, 20% EtOH) at 90 V for 3 h at 4 °C via wet transfer. Membranes were stained with Ponceau S, scanned and destained with TBS-T (50 mM Tris-HCl pH 7.5, 150 mM NaCl, 0.1% [v/v] Tween-20). Membranes were then blocked in blocking buffer (5% [w/v] skim milk powder and 0.01% [w/v] sodium azide in PBS) at room temperature for 1 h, incubated in primary antibodies diluted in PBS (1:1000 [v/v]; 1:500 [v/v] for anti-phospho H3 (Ser10)) containing 0.01% [w/v] sodium azide overnight and washed 5 times with PBS. Membranes were then incubated with horseradish peroxidase (HRP)-conjugated secondary antibodies (Jackson Immunoresearch, diluted 1:5,000-10,000 [v/v] in PBS). For detection, membranes were shortly incubated in Western Lightning Enhanced Chemiluminescence substrate (PerkinElmer) and imaged with a Vilber imaging system.

## Polysome profile analysis and fractionation

$1 \times 10^6$ HeLa cells were seeded 12 h before starting the experiment and kept subconfluent to avoid translation attenuation by contact inhibition. For siRNA transfections, $4 \times 10^5$ HeLa cells were reverse transfected for 24 h if not indicated differently. CHX (100 µg/ml) was added to cells for 5 min at room temperature before washing with ice-cold PBS. Cells were then scraped in polysome lysis buffer (20 mM Tris-HCl pH 7.5, 150 mM NaCl, 5 mM MgCl$_2$, 1 mM DTT, 100 mg/ml CHX, 1% Triton X-100, 40 U/ml RNasin, and EDTA-free protease inhibitors (Complete Mini, Roche)); 100 µl polysome lysis buffer were used for $4 \times 10^5$ HeLa cells, 150 µl for $1 \times 10^6$ HeLa cells. The lysates were rotated for 10 min at 4 °C and cell debris was pelleted by centrifugation at $10,000 \times g$ for 10 min at 4 °C. For sucrose density gradient centrifugation, the cytoplasmic lysates were carefully loaded onto linear 17.5–50% [w/v] sucrose gradients (dissolved in 20 mM Tris-HCl pH 7.5, 5 mM MgCl$_2$, and 150 mM NaCl) and centrifuged in an SW60 rotor (Beckman) for 2.5 h at 40,000 rpm (swing bucket rotor, RCF$_{max}$ = 208,600 × g; RCF$_{average}$ = 157,600 × g) and 4 °C. Polysome profiles were recorded using the absorption at 254 nm with a Teledyne ISCO Foxy Jr. or a Teledyne ISCO Foxy R1 system coupled with the PeakTrak software. The profiles were aligned, normalized, and quantified using the QuAPPro App[36]. For calculating the percentage of 40S, 60S, 80S and polysome peaks, the area under the respective peaks was divided by the total area under the curve.

For the fractionation of sucrose density gradients, approximately 300 µl eluate per fraction was collected every 14 s during gradient elution. For recovery and concentration of protein from the fractions, 10 µl Strataclean Resin beads (Agilent Technologies) in 300 µl 20 mM Tris-HCl (pH 7.5) were added to each fraction. The fractions were rotated end-over-end at 4 °C overnight and centrifuged at 211 × g for 2 min. After discarding the supernatants, protein was eluted from the beads by addition of 20 µl 2 × SDS Laemmli buffer and heating at 95 °C for 10 min. 14 µl of each sample were loaded on 12% SDS-polyacrylamide gels for Western blot analysis.

## RT-qPCR analysis

Total RNA was isolated and on-column DNase digested using the Universal RNA Purification Kit (EURx/Roboklon). For cDNA synthesis, 1 µg total RNA was reversed transcribed with 100 U M-MLV Reverse Transcriptase (Promega), 4 µM Random Hexamer Primers (Thermo Fisher Scientific), 20 U RNasin Ribonuclease Inhibitor (Promega) and dNTPs (1 mM each) for 1 h at 37 °C. PCR reactions (10 µl) were mixed in 384-well plates using a 5-fold dilution of the cDNA reaction, 400 nM of

each gene-specific primer and the Powerup SYBR Green Master Mix (Thermo Fisher Scientific). The Quantstudio5 system (Thermo Fisher Scientific) was used for quantitative PCR. For every target/reference gene, triplicates were measured. All measurements included negative controls lacking reverse transcriptase to assess genomic DNA contamination, and the efficiency of each primer pair was determined by a cDNA dilution series. For mRNA detection, gene-specific primers were designed with help of the Universal Probe Library Assay Design Center (Roche), and synthesized by Eurofins Genomics as follows:

G5085, 5'-CCAACCGCGAGAAGATGA-3' (ACTB fwd)
G5086, 5'-CCAGAGGCGTACAGGGATAG-3' (ACTB rev)
G5767, 5'-CGGCAAGATGGCAGTGCAAA-3' (RPS3 fwd)
G5768, 5'-TTCAGCCAGCTCCCGAGTAA-3' (RPS3 rev)
G5769, 5'-GCAACTGTGGTGGAGATTGC-3' (RNF10 fwd)
G5770, 5'-CTCACAGGTGAGTGGCAAGT-3' (RNF10 rev)

## Antibodies

The following antibodies were used: Rabbit anti-RPS6 (5G10; Cell Signaling, 2217), rabbit anti-RPS3 (Bethyl, A303-840A), rabbit anti-RPS19 (Bethyl, A304-002A), rabbit anti-RPS20 (Abcam, ab133776), rabbit anti-RNF10 (Proteintech, 16936-1-AP), rabbit anti-ZNF598 (Sigma, HPA041760), mouse anti-Tubulin (Abcam, ab6160), rabbit anti-RPL7 (Proteintech, 14583-1-AP), rabbit anti-RPL5 (Cell signaling, 14568), mouse anti-RPL11 (Thermo Fisher Scientific, 37-3000), mouse anti-p53 (DO-1, Santa Cruz), mouse anti-FLAG (M2, Sigma, F3165), rabbit anti-EDF1 (Abcam, ab174651), rabbit anti-phospho-histone H3 (Ser10) (Cell Signaling, 9701), rabbit anti-histone H3 (Abcam, ab1791), mouse anti-puromycin (Millipore, MABE343), and rabbit anti-ubiquitin-K48 (Abcam, ab140601).

## Puromycin incorporation assay

HeLa cells were treated with DMSO (negative control) or ANI at the indicated concentrations for 2 h. Puromycin (10 ug/ml, Applichem) was added to the culture medium 10 min before harvesting the cells. The amount of puromycin incorporated into nascent polypeptides was measured by Western blot analysis using anti-puromycin antibody (1:1000 [v/v]), and normalized to the amount of total protein stained with Ponceau S (Applichem).

## Proliferation assay

To test the impact of RNF10 on cell proliferation, $1 \times 10^5$ HeLa cells were seeded in 6-well plates and grown over a period of 4 days in the absence or presence of low dose ANI (0.02 µg/ml). After 24 h, 48 h, and 72 h, cells were trypsinized, resuspended in 1 × PBS and counted in a Newbauer chamber. The proliferation rate was calculated by fitting an exponential curve to the data points using Excel. To obtain relative proliferation rates, each proliferation rate was normalized to the average proliferation rate of untreated HeLa cells.

## RQC reporter gene assay and FACS analysis

For siRNA KD and transient expression of the GFP-mCherry double fluorescence ribosome stalling reporter, $2 \times 10^5$ HeLa cells were reverse transfected with 100 nM siRNA according to the manufacturer´s instructions with RNAiMAX Lipofectamine transfection reagent (Thermo Fisher Scientific) for 14 h followed by incubation in fresh medium for 6 h. Cells were then transfected with 2 µg of the control reporter plasmid pmGFP-P2A-3xFLAG-(K)$_0$-P2A-mCherry (p3878) or the ribosome stalling reporter plasmid pmGFP-P2A-3xFLAG-(K$^{AAA}$)$_{20}$-P2A-mCherry (p3880)[37] for 6 h using the TurboFectin 8.0 (OriGene) transfection reagent according to the manufacturer's protocol, followed by 48 h incubation in fresh medium.

For RQC reporter gene analysis, cells were trypsinized, resuspended in 1% [v/v] FBS/1 × PBS, washed with 1 × PBS, and resuspended in FACS buffer (10 mM HEPES, pH 7.2; 0.5 mM EDTA, 2% BSA) and 1.25 nM Sytox Red Dead Cell Stain (Invitrogen, S34859). FACS analysis

was performed at the FACSAria Illu (BD Fusion Cell Sorter; FlowCore, Medical Faculty Mannheim) with the BD FACS Diva software v9.0.1, and analyzed with the FlowJo software v7.6.5. GFP and mCherry positive cells were gated manually. For quantitative evaluation of RQC efficiency, the median values of the GFP and mCherry populations of 11,000-29,000 GFP positive cells were used for calculating the mCherry/GFP ratio.

## Statistics and reproducibility
All data are presented as mean ± SD if not indicated otherwise. Experiments were repeated independently, the number or repeats (*n*) and the type of statistical test is indicated in the figure legends. Western Blot signals were quantified using ImageJ v1.52[53]. GraphPad Prism v8.4.1 and Microsoft Excel 2019 was used for statistical analyses.

## Reporting summary
Further information on research design is available in the Nature Portfolio Reporting Summary linked to this article.

## Data availability
All data are available within the article, Supplementary Information or Source Data file. Source data are provided with this paper.

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

## Acknowledgements

We would like to thank Alessia Ruggieri (Center for Integrative Infectious Disease Research, University Hospital Heidelberg) for kindly providing plasmids for lentiviral transduction, Stefanie Uhlig (FlowCore Mannheim, Medical Faculty Mannheim of Heidelberg University) for cell sorting, Adelheid Cerwenka (Mannheim Institute for Innate Immunoscience, Heidelberg University, Medical Faculty Mannheim, Germany) for kindly providing time at the BD Fusion Cell Sorter, Bogdan Jovanovic (MI3, Heidelberg University) for generating plasmid p3437, Ramanujan Hegde (MRC Laboratory of Molecular Biology, Cambridge, UK) for kindly providing the ribosome stalling reporter plasmids, as well as Chiara Schiller and Johanna Schott (MI3, Heidelberg University) for developing the QuAPPro App for interactive alignment, quantification and visualization of polysome profiles. This work was funded by the Deutsche Forschungsgemeinschaft (DFG) project numbers 278001972—TRR 186, 439669440—TRR 319, 464424253—CRC 1550 and 445549683—RTG 2727 to G.S.

## Author contributions

G.S. and J.A.L. designed the study, analyzed data and wrote the manuscript. J.A.L. carried out the majority of experiments. D.L. assisted with plasmid cloning and lentiviral particle production. H.M.S. conducted puromycin incorporation experiments.

## Funding

## Competing interests

The authors declare no competing interests.
