## [Transparent Peer Review file · Nature Communications]

E3 ubiquitin ligase RNF10 promotes dissociation of stalled ribosomes and responds to ribosomal subunit imbalance

Corresponding Author: Professor Georg Stoecklin

Version 0:

Reviewer comments:

Reviewer #1

(Remarks to the Author)

Lehmann et al describe effects of RNF10-mediated RPS3 ubiquitination upon treatment with (1) translation elongation inhibitors and (2) under conditions of ribosome biogenesis perturbation. While most of the results for point (1) have previously been published (Garzia et al 2021 and Garshott et al 2021), there is some new characterization of the action of RNF10 in resolving half-mer ribosome populations under these conditions. Additionally, ribosome biogenesis perturbation (hitting both the large and small subunit) appears to impact RNF10 levels in a reciprocal fashion suggesting a coupling of these processes. Finally, the authors show that RNF10 overexpression leads to decreased levels of polysomal half-mers upon knockdown of 60S proteins.

The new findings of this paper are the links between RP levels (biogenesis) and RNF10 levels, as well as the impact of RNF10 overexpression on half-mer formation. However, RNF10-mediated ubiquitination of RPS3 has been previously characterized both in yeast and in mammals thus detracting from the novelty of the study. The new observations are generally interesting, but the depth of insight is limited and several key questions are underexplored. For instance, the half-mers formed in distinct conditions of elongation inhibition and RPL knockdown likely reflect partially resolved collisions or iRQC complexes, respectively (where the 40S subunit is either lagging or leading in the complex). The new data suggest that these two different complexes may both be processed by RNF10. How do different half-mer complexes (resulting from RPL KD or elongation inhibition) get cleared? Moreover, how significant is RNF10's endogenous role? Many of the results rely on overexpression of RNF10 and thus the significance of RNF10's endogenous role is rather unclear.

Overall, this a clearly written manuscript on an interesting topic with some strong experimental data, and some substantial overlap with previous work. There are also some new observations about RNF10 regulation though these are relatively under characterized and only discussed speculatively in the discussion.

Major comments

1. Many of the results have been shown by previous publications, including Garzia et al, 2021 and Garshott et al, 2021. While these publications are mentioned, their citation is lacking throughout, especially where figures show the same results that have been published. The contributions of published papers should be clearly acknowledged throughout. Below are results from this manuscript that have been previously published:
 - In Fig. 1A, the effect of elongation inhibitors on RPS3-Ub has been shown by Garzia et al, 2021.
 - Similar results to Fig. 1C have been shown by Garzia et al and Garshott et al. While I appreciate that anisomycin treatment is a (slightly) different condition from published results, ubiquitination of RPS3 in ribosomal fractions has been shown and this should be acknowledged.
 - The findings of Fig. 1D are not new, as Garzia et al published that K214 is the residue on RPS3 that is ubiquitinated.
 - Garzia et al should be cited for the C225S/C228S mutant of RNF10.
 - In Fig. 2, increased Ub-RPS3 and increased free 60S subunits upon RNF10 overexpression were observed by Garzia et al, as was the lack of phenotype of RNF10 KO.
 - In the discussion of Fig. 6 in the text, there needs to be citation of Garzia et al. in finding that RNF10 activity responds to initiation and elongation stalls.
2. RNF10 KO does not alter many of the phenotypes explored in this manuscript. While RNF10 KO prevents RPS3 ubiquitination upon translation elongation inhibition (Fig. 1), this does not alter half-mer formation, 40S subunit levels, or global translation more generally, as assessed by polysome profiling. Most of the strong phenotypes of translation dynamics

characterized here only occur upon very strong RNF10 overexpression. This may suggest that at endogenous levels, RNF10's contribution to translation and/or ribosome dynamics is minor. To ask whether RNF10 is a major factor in the response to elongation stress or 60S biogenesis perturbation, the authors could test whether RNF10 KO impacts protein output or ribosomes levels over a longer period of stress. At the very least, the authors should discuss what the lack of phenotype in RNF10 KO cells may mean for how endogenous levels of RNF10 contribute to ribosome homeostasis.

3. Several points made by the new data in this manuscript are underexplored.

- The correlation of RNF10 levels with 40S/60S levels is interesting, but RNF10 degradation is not explored beyond proteasome inhibition.

- i. How rapidly is RNF10 degraded upon knockdown of 40S proteins? Does this correlate with 40S protein levels? A time course of these protein levels upon 40S protein knockdown would be useful.

- ii. Upon 60S protein knockdown, are increased RNF10 levels directly dependent on further stabilization? Testing whether proteasome inhibition impacts RNF10 levels +/- RPL knockdown could help parse whether RNF10 levels are impacted by increased stability or potential increased protein synthesis.

- iii. The mechanism of RNF10 degradation could be clarified using a diGly proteomics approach to identify ubiquitinated residues or screening a library of siRNAs targeting E3 ligases to clarify how RNF10 is targeted to the proteasome.

- It is interesting that RNF10 overexpression can decrease half-mers both upon elongation inhibition and RPL knockdown. How RNF10 clears these distinct complexes is unclear.

- i. Upon elongation inhibition, Ub-RS3 is associated with polysomes but not with the free protein fraction (Fig. 1c); upon RPL7 knockdown, the converse is true (Fig. 4a). Could the authors comment on how this may relate to the mechanism of RNF10-mediated ubiquitination in these different conditions?

- ii. Additionally, upon 60S biogenesis perturbation, it is not completely clear whether RNF10 is ubiquitinating RPS3 associated with free 40S subunits or 40S subunits stalled on mRNAs. Ub-RPS3 is not associated with polysomal half-mers upon RPL7 KD (Fig. 4a), yet RNF10 overexpression clears these polysomal half-mers (Fig. 6c) – how do the authors parse this discrepancy? The authors should test whether RPL7 KD with RNF10 overexpression induces Ub-RPS3 associated with polysomal half-mers.

4. Some controls are lacking. For example, in Fig. 3, the accounting of protein levels after siRNA KD should be more careful. RPS3 protein levels from whole cell lysate shown be shown for Fig. 3A/B. Indeed, KD of RPS3 does not appear to have a significant impact on RPS3 levels in Fig. 3C. It is understandable that KD of a ribosomal protein would not yield cells with a very strong knockdown, but it warrants comment. Quantification of RP levels upon KD would be helpful to understand the degree of knockdown.

5. As a final point, some of the references selected for background material were surprising and not accurate. For example, the reference chosen for NGD (15) refers to work in bacteria rather than to the abundant work on this topic in eukaryotes.

Other comments

1. In Fig. S1B, could the authors comment on why RNF10 levels decrease upon ZNF598 KD?

2. In Fig. S2C, RNF10 overexpression does not appear to increase free 60S subunits as it does in other figures. Could the authors address this inconsistency?

3. KD of some small subunit proteins appears to impact levels of other small subunit proteins (see Fig. 3C,E) and this is also true for large subunit proteins (see Fig. 4D,F). This likely points to disruptions in small and large subunit stability and is an interesting point that warrants comment.

4. In Fig. S3D, could the authors speculate as to why siRPS6 and siRPS20 cause differential effects on polysome profiles?

5. In Fig. S3E, the implication is that RIO2 KD is causing decreased Ub-RPS3 via decreased RNF10 levels. An alternative explanation could be that overall RPS3 levels are decreased upon RIO2 KD, leading to decreased Ub-RPS3. Quantification of RPS3 and Ub-RPS3 levels in this experiment could help clarify this point.

6. Fig. 5 is not very relevant to the main conclusions of the paper and should be moved to the supplement.

- In Fig. 5D, can the authors comment on why ZNF598 levels are decreased upon treatment with any siRNA other than the control siRNA?

Reviewer #2

(Remarks to the Author)

Review for the manuscript NCOMMS-22-43495 by Dr. Stoecklin and co-authors entitled "The E3 ubiquitin ligase RNF10 promotes dissociation of stalled ribosomes and responds to ribosomal subunit imbalance"

It has been demonstrated that the ubiquitination of uS3 is essential for the quality control of 18S nonfunctional rRNA decay (NRD) that induces the degradation of the nonfunctional ribosome with the mutation in the decoding center. In yeast, 18S NRD depends on the polyubiquitination of uS3 by a set of ubiquitination enzymes. Mag2 is essential for 18S NRD, and responsible for the monoubiquitination followed by the polyubiquitination. Fap1 is the main E3 ligase for 18S NRD and Hel2 partially contributes to the degradation of the nonfunctional ribosome. Although the 18S NRD in higher eukaryotes is largely unknown, however, the involvement of RNF10, a human homolog of Mag2 in the degradation of the 40S subunit has been suggested. RNF10 has an activity for the monoubiquitination of uS3 at K214, the compatible residue of the yeast uS3 K212. However, it is unknown whether the monoubiquitination is followed by polyubiquitination with ZNF598 or other E3 ligases in higher eukaryotes.

In this study, the authors demonstrated that translation elongation stalling induces mono-ubiquitination of RPS3 at K214 by RNF10. RNF10 overproduction reduces the halfmers formation induced by anisomycin treatment and induced by RPL KD condition. The proposal is potentially interesting. However, the mechanism by which RNF10 reduces the halfmers is not addressed. Although RNF10 overproduction reduces the halfmer formation induced by anisomycin treatment, the RNF10-

KO phenotype is not evident. Moreover, it is unclear whether the reduction of the halfmers is due to the dissociation of 40S by RNF10 simply because the dissociation of the 40S subunit from halfmers is not experimentally demonstrated. It is no experimental analysis to examine how the RNF10 level is reduced in a proteasome-dependent manner in response to the KD of RPS proteins, and how the RNF10 level is increased in response to the KD of RPL proteins. The relation between proteasomal degradation of RNF10 upon RPS-KD and upregulation of RNF10 upon RPS-KD is unaddressed. There is no mechanistic insight for the regulation but transcription factor p53 does not control RNF10 expression. Overall, the experiments that support the proposal are largely missing, more experimental evidence must be done to support the activity of RNF10 proposed in this study. It's reasonable to revise the overstatement in the discussion based on the experimental results shown in this manuscript.

Major points

1. The authors proposed that the mono-ubiquitination RPS3 K214 by RNF10 upon translation elongation stalling is involved in RNF10-mediated rescue and suppression of ribosomal halfmer formation. This assumption should be directly investigated by the experiments using RPS3-K214R mutant cells. The authors demonstrated that RNF10 overproduction reduces the halfmer formation induced by anisomycin treatment (Figure 2b) and induced by RPL KD condition (Figure 6c). The phenotype of the RPS3-K214R mutant cells will strongly support that the ubiquitination of uS3 is required for the RNF10-mediated reduction of the halfmer.
2. It is unclear whether the reduction of the halfmers is due to the dissociation of 40S by RNF10 simply because the dissociation of the 40S subunit from halfmers is not experimentally demonstrated. In the related RQC filed, *in vitro* reconstitution of ZNF598-mediated ribosome ubiquitination and subunit dissociation by the RQT complex have been reported (PMID: 32579943 PMID: 36302773). I highly recommend, to conclude RNF10 promotes the dissociation of stalled ribosomes shown in the title, the RNF10-mediated ubiquitination of the halfmers and subunit dissociation should be demonstrated by the *in vitro* reconstitution reaction.
3. In the previous report (PMID: 343481610), ZNF598-KO reduces the level of mono-ubiquitination of uS3 with the anisomycin treatment. To make clear the difference, the levels of monoubiquitinated uS3 in Hela-WT and ZNF598-KO cells with the concentration of anisomycin should be compared.

Reviewer #3

(Remarks to the Author)

Lehmann et al. provide evidence for a feedback loop between 40S ribosomal subunit levels and RNF10. Translation and RNF10 are disrupted using multiple orthogonal techniques, including drugs, knockdown, and overexpression. Ribosomal state is measured by monitoring ubiquitination and translation state via sucrose gradients. The authors speculate on the mechanism by which RNF10 acts upon the small ribosomal subunit but do not offer direct evidence. This study complements previous work showing that proteostasis stressors regulate 40S levels (PMID: 34469731). This is a valuable systems study of how cells maintain ribosome levels in dynamic and stressful conditions. It suggests a translation-sensitive feedback loop that may be orthogonal to better understood mechanisms such as those involving ZNF598. I recommend publication after minor changes described detailed below.

Line 246: The authors mention half-mer formation and reference figs 4a and 6a. Yet these figures panels do not clearly highlight or quantify (better) half-mers as is done in 2b or 6c. Did the authors mean to reference 6c? Figures should be updated to highlight half-mer levels.

Fig 1c: Any comment on why there are increased levels of free RPS3 when RNF10 is knocked out? Is RNF10 leading to ubiquitination and degradation of unassembled RPS3?

Fig2a: mutant RNF10 is destabilized vs WT. Please address in text.

Discussion section: A cartoon would be useful for summarizing the findings of this work and communicating potential molecular mechanisms.

Reviewer #4

(Remarks to the Author)

The E3 ubiquitin ligase RNF10 promotes dissociation of stalled ribosomes and responds to ribosomal subunit imbalance submitted by Lehmann et al. focuses on the role of RNF10 in RPS3 monoubiquitylation and its relation to ribosomal subunit stoichiometry regulation.

This manuscript presents results from extensive cellular studies that in addition provided basis for analysis of mRNA levels and polysome profiling. These experiments uncovered interesting relationship between levels of 40S/60S subunits and RNF10 and show that RNF10 is responsible for clearance of 40S subunit via RPS3 ubiquitylation at K214.

Manuscript is overall well written and nicely organized with a good flow. Several suggestions were made below to help improve the logic of the story and overall understanding of the text. Considering that authors address these comments, I am happy to recommend this manuscript for publication.

General comments:

Authors in the introduction describe RNF10 as protein characterised in the nervous system, however, in the subsequent result sections use cell lines derived from cervical, retinal, and colorectal cell lines. Additionally, considering information available in the Human Atlas Database (<https://www.proteinatlas.org/ENSG00000022840-RNF10/tissue>) and the wide expression pattern, it would be helpful to understand the logic behind the cell line choices made. If there was preceding cell line screening, it might be helpful to disclose this information.

Considering that the authors focus on the monoubiquitylation of RPS3 it would be helpful to include a representative western blot for each cell line showing the whole membrane in the supplementary materials. Highly active E3 ligases tend to form polyUb chains that are usually resolved between the higher MW regions and the stacker. If there are any findings, it would be also interesting to bring this into the context with the two-step mechanism proposed for Mag2 polyubiquitylation.

I am wondering if it is possible for the reader to compare individual polysome profiling figures. Are the ranges on Y-axes of relative units for the absorbance at 254 nm in the same range? Was the data normalised in any way?

Specific comments:

Authors mentioned structural interface between small subunits (ref. 18). Considering this publication and additional available structural data, two question arise for me:

1. It appears that S3 proteins and their K212 residues are in a very close proximity at the interface. Do authors think that this proximity plays any role in the regulation of Ub attachment?
2. Considering that authors probed RPS3, RPS6, RPS19, and RPS20 and their effect on RNF10 levels, I wonder how the choice of these particular structural proteins was made. Interestingly, upon mapping these on the available structures, RPS20 seems to contact RPS3. Additionally, similar question arises for the large subunit RPL protein selection.

ZNF598 silencing does not seem to be as efficient as the one for RNF10. It might be helpful to quantify figures such as S1e, where there seems to be difference between Ub1 levels. Do authors think there might be any residual activity? Additionally, it is interesting that the RNF10 levels in Fig. 1a (lanes 1,2,5,6) and S1b (lanes 1,2,3,4) seem to be different in relation to the control. Considering that the same siRNA S193 was used, can authors explain this? Is there any possibility in any way that ZNF598 influences RNF10 levels?

In the relation to the line 111, in the Fig. 1c authors comment that there is no monoUb in the fraction #1. It appears that there is a weak band which seems very minor. However, it is interesting that the Ub1 levels seem to fluctuate with the same trend as RPS3 levels. It would be helpful to compare these ratios by quantification and accordingly adjust narrative if needed.

Minor comments

Line 62

Further comment might be needed to avoid confusion as Hel2 in humans is a 14-3-3 type protein.

Line 65

Was RNF10 the only outstanding candidate? If no, it might be helpful to provide an alignment figure and logic why only RNF10 was tested.

Line 184

RIO2 is introduced here without any previous context. Can authors explain why they probed this protein in particular?

Line 237

Considering that authors performed quantification in Fig. 5b by RNF10 ratio to Tubulin, I wonder whether the measured values for Tubulin in Fig. 5a remained in the same range. Visually it appears that these levels fluctuate.

Lines 240 & 513

Fig 5.d includes western blot for p21. It would be helpful to explain this in the text.

Lanes 472 & 483

Although the rotor type is mentioned here, it would be helpful to unify the units used during centrifugation to RCF.

Version 1:

Reviewer comments:

Reviewer #1

(Remarks to the Author)

The authors have done a thorough job addressing reviewer comments and I am now enthusiastic about publication. Below are listed a few lingering questions that are NOT essential to address for publication.

1. The revised manuscript more thoroughly cites published work that demonstrated RNF10-mediated ubiquitination of RPS3 in response to elongation/initiation inhibitors.
 - Garzia et al 2021 and Garshott et al 2021 should be cited in the first mention of Fig. 1a. I appreciate that the authors have demonstrated RPS3 ubiquitination in response to a wider variety of elongation inhibitors, but both these papers previously showed the same result with anisomycin and should be acknowledged alongside the data presented here.
2. The authors have made efforts to test phenotypes associated with RNF10 KO. They find that RNF10 KO causes minor increases in halfmer formation with low dose anisomycin treatment. RNF10 KO cells also have slightly impaired proliferation under basal conditions that becomes more pronounced under low dose anisomycin.
 - It would be useful to show the absolute proliferation rates for untreated and anisomycin-treated cells.
 - Could the authors comment on why protein synthesis (as measured in their puromycin incorporation assay) does not appear to decrease with anisomycin treatment, despite anisomycin significantly impairing proliferation? Further, it seems odd that RNF10 expression affects proliferation but not protein synthesis – could the authors comment on this?
3. The authors have made some effort to explore RNF10 degradation using the proteasome inhibitor MRZ.
 - They propose that smearing above the RNF10 band upon MRZ treatment may reflect ubiquitination of RNF10. To test this, they could IP their tagged RNF10 constructs and probe them using a ubiquitin antibody.
 - The authors comment that MRZ treatment is toxic to cells after 8 hours and prevents RPS3 ubiquitination. Is MRZ treatment inhibiting translation? This could be tested with polysome profiles of MRZ-treated cells and may explain why RPS3 is not ubiquitinated if translation initiation is inhibited by MRZ.
 - The MRZ data argue that further stabilization of RNF10 does not explain increased RNF10 levels with RPL knockdown. The authors could test whether translation of RNF10 is increased upon RPL knockdown by checking whether the amount of RNF10 mRNA in polysomes increases (or not) upon RPL knockdown.

Reviewer #2

(Remarks to the Author)

Review for NCOMMS-22-43495A

I appreciate the author's efforts to prove their proposal. However, I want to highlight that in the revised manuscript, there are still several overstatements without direct evidence to support the claim of 'the dissociation,' such as "The E3 ubiquitin ligase RNF10 promotes dissociation of stalled ribosomes" in the title and "RPS3 ubiquitination at K214 promotes dissociation of stalled 40S subunits" in Figure 3 legend. It's challenging for me to recommend the publication without the correction of these overstatements.

Reviewer #4

(Remarks to the Author)

Lehmann et al. submitted a revised manuscript characterizing E3 ubiquitin ligase RNF10 and its relation to the stalled ribosomes during translation.

Additional experiments helped to clarify some of the conclusions and link the related sections. I appreciated the addition of the final diagram and also the figures in the comments to the reviewers. Some of these, as well as the tables, would be helpful for the main text too.

Several additional comments have been made below.

General Comments

I found the magnification of the polysome profiling very helpful. I would suggest to include more of them, especially in Fig. S2bc, S5cd. Additionally, some of the profiles exhibit dips in the absorbance (Fig. 1 or Fig. S5). It would be good to comment on this observation. Lastly, the authors mention free fraction of the profiles in the text. It would be good to mark this on the X-axis.

In the experiments presented in Fig. 4 and 5, multiple RPS and RPL were silenced. This appears to have a sequential effect where the silencing of one protein leads to reduced levels of the other analysed proteins in the subunit. It would be very helpful to provide some additional comments or accompany these blots with quantification. Does this silencing trigger a cascade effect where the whole subunit is being cleared? Additionally, the silencing of the 40S proteins effect on 60S levels visually appears to be stronger than vice versa. I wonder if the data can be interpreted and thus contribute to the conclusions and model authors presented.

Major comments

Authors have defined RQC as a process that includes the degradation of the nascent peptide, ribosomal proteins, and affected mRNA. Based on this definition, if the ribosomes stop due to the mRNA-related effects, efficient RQC should result in the clearance of these elements and restart of the translation.

Considering that the polyLysine reporter induced stalling, but some mCherry signal is still observable, a percentage of the translation runs into completion. Wouldn't this suggest that KD of ZNF598 results in higher clearance of the stalled components and thus re-enabling the system to carry out translation and thus increase mCherry production? If so, then the statement in Line 172 about reduced RQC efficiency seems confusing. I suggest defining RQC efficiency and rewording this narrative to make the characterized process easier to understand. Additionally, explaining the link between RQC and RQT would be helpful. Similar could be done for ZNF598 and RNF10 to make the hypothesis clearer. Overall, this section (Lines 164-190) introduces many elements with some logical links missing.

Lines 293-296/600 and Fig. S5f – This passage is somewhat confusing. The authors show the figure with the Ub-K48 label, however, the K48 linkage-specific antibody is not mentioned in the methods. Does this panel represent polyubiquitinated RPS3? This is also not described in the text. Furthermore, do authors by recognition mean proteasomal recognition? Canonically, K48-linked polyubiquitin chains of four or more ubiquitin molecules are required for efficient proteasomal degradation, although degradation of monoubiquitylated targets is also possible. This section of the text needs to clarify whether the observed event is related to degradation or just recycling, and the type of modification that was observed.

Minor comments

Fig. S1g – KO-control is not described. Does this represent KO by CRISPR/Cas9 without gRNA?

Line 103 – Authors claim that ZNF598 KD had no effect on Ub1-RPS3 levels, however, siRNA 201 coupled with ANI and CHX treatment appear to have lower band intensity. Quantification would help to make this clearer.

Line 172/621/622/Fig. S2e – reporter naming. It would be good to unify nomenclature. I would also suggest omitting AAA usage as the mixing of amino acid and nucleotide codes is confusing and suggests a Lysine/Alanine sequence.

Fig. 3a – K214A shows a minor Ub1 band upon ANI treatment which is comparable with the WT in the RPS3 panel but not in the FLAG panel. Can the authors please explain this discrepancy? Considering this experiment is done pre-silencing, do authors think that there might be a lower efficiency of FLAG-RPS3 incorporation into the ribosomal subunit, thus generating two populations? One way of testing this would be to show Ub1-RPS3 levels from the profiling fractions and compare free vs. 40S levels, similar to Fig. 1c

Line 244 – How were the RPS20 levels of 50% calculated/quantified?

Line 287 – figure number missing

Version 2:

Reviewer comments:

Reviewer #1

(Remarks to the Author)

The authors have addressed my concerns.

Reviewer #4

(Remarks to the Author)

Lehmann et al. have done well addressing the comments of the reviewers and taking time to explain the details, along with bringing figures to a more understandable level. In summary, I feel positive to suggest this manuscript for publication.

Changes in the manuscript

- **Fig. 1:** In panel c, we show a new polysome profile analysis using two pools of RNF10 KO cells rather than individual clones. Western blots for EDF1 were also added to this analysis. Fig. 1d represents a new quantification of the RPS3 monoubiquitination signal across the polysome profiles. The Western blot of former Fig. 1d was replaced and is now shown in Fig. 3a.
- **Suppl. Fig. 1:** Panel a shows a new quantification of RNF10 expression from Western blots in Fig. 1a. In Suppl. Fig. 1c, the Western blot of former Suppl. Fig. 1b was replaced with a new repeat experiment, documenting that RNF10 levels are elevated upon ZNF598 KD in RPE1 cells. A quantification of this effect is also shown in Suppl. Fig. 1d. Suppl. Fig. 1e was updated from former Suppl. Fig. 1c. In Suppl. Fig. 1g, a new Western blot was added showing RNF10 and RPS3 expression in the two RNF10 KO pools. Suppl. Fig. 1i depicts new data on expression levels of RNF10 and RPS3 monoubiquitination after extended KD of ZNF598 in HeLa cells using two consecutive siRNA transfections.
- **Fig. 2:** The Western blot in panel a was replaced, showing that anisomycin does not cause a change in the expression of transgenic RNF10. In Fig. 2c, new polysome profiles are shown from cells grown at low concentrations of anisomycin (0.02 $\mu\text{g/ml}$) for extended periods of time (48 h). Fig. 2d depicts a new scheme illustrating how ribosomal half-mers were quantified. Fig. 2e shows the quantification of ribosomal half-mers from Fig. 2c, demonstrating an increase in RNF10-KO cells, and a decrease in cells overexpressing RNF10-WT. New data are shown in Fig. 2f from a proliferation assay in the presence of low dose ANI, demonstrating a pronounced proliferation defect in RNF10-KO cells under conditions of chronic translation elongation stress. The quantification shown in former Fig. 2c was moved to Suppl. Fig. 2a.
- **Suppl. Fig. 2:** In Suppl. Fig. 2d, which replaces former Suppl. Fig. 2c, we show new polysome profiles after temperature adjustment of the detector to improve reproducibility of the profiles. As in our other experiments, we observe an increase of free 60S subunits upon RNF10-WT overexpression. In Suppl. Fig. 2e, new results from a FACS-based RQC reporter system are shown, demonstrating that RNF10 does not affect RQC activity.
- **Fig. 3** is a new figure. Panel a shows a new Western blot from Flag-RPS3-WT/K214A expressing HeLa cells, with more equal loading, replacing the blot in former Fig. 1d. New data are presented in Fig. 3b on the difference in ribosomal half-mer formation between Flag-RPS3-WT and Flag-RPS3-K214A expressing cells. In Fig. 3c, free 40S subunits and half-mers were quantified from Fig. 3b, demonstrating that RPS3-K214 is important for dissociation and turnover of stalled 40S subunits.
- **Suppl. Fig. 3** shows polysome profiles that serve as controls to the new experiment of Fig. 3b.
- **Fig. 4** replaces former Fig. 3. Panels b and e of former Fig. 3 were moved to Suppl. Fig. 4a and f, respectively, to make space for new data. A new time course experiment showing the kinetics of RNF10 degradation upon RPS3 and RPS20 KD is shown in Fig. 4d. Western blots showing RPS3, RPS20 and RPS6 were added to Fig. 4e (former Fig. 3g). Former Fig. 3f on the effect of RIO2 KD was omitted from the manuscript to streamline the story.
- **Suppl. Fig. 4** replaces former Suppl. Fig. 3. A Western blot showing RPS3 was added to Suppl. Fig. 4a, and graphs from quantification of RPS3 and Ub₁-RPS3 levels were added to Suppl. Fig. 4b. In Suppl. Fig. 4g, we show new polysome profiles, replacing those from former Suppl. Fig. 3d, after temperature adjustment of the detector, which improves reproducibility of the profiles. In Suppl. Fig. 4h, we added new polysome profiles after RPS19 KD. Former Suppl. Fig. 3e and f on RIO2 KD were omitted from the manuscript to streamline the story.
- **Fig. 5** replaces former Fig. 4. In Fig. 5g, we now added a quantification of the Ub₁-RPS3 distribution across polysome profiles in RPL7 DK versus ANI-treated cells.
- **Suppl. Fig. 5** replaces former Suppl. Fig. 4. Panel b contains a new quantification of ZNF598 levels from Fig. 5b (former Fig. 4b). New Suppl. Fig. 5d shows Western blots after polysome fractionation for RPS3 and EDF1, comparing RPL7 KD and ANI-treated cells. Quantification of the Ub₁-RPS3/RPS3 ratio from these blots is depicted in Fig. 5g. Suppl. Fig. 5e-g shows new data

concerning the effect of proteasome inhibition on RNF10 protein levels upon RPL7, RPL11 and RPL 5 KD.

- **Fig. 6** replaces former Fig. 5. Panel c from former Fig. 5 was moved to Suppl. Fig. 6a. Panel d from former Fig. 5 was omitted from the manuscript.
- **Suppl. Fig. 6:** Panel b was added, showing the quantification of RNF10 levels from the Western blot in Suppl. Fig. 6a (former Fig. 5c).
- **Fig. 7** replaces former Fig. 6. Panels a and b from former Fig. 6 were moved to Suppl. Fig. 7a and b. Quantification of half-mers in Fig. 7a is now shown as a new panel in Fig. 7b. Former Suppl. Fig. 5 was moved to Fig. 7c.
- **Suppl. Fig. 7** replaces former Suppl. Fig. 5. Panels a and b were moved to the supplement from former Fig. 6a and b. Quantification of half-mers in Suppl. Fig. 7a was added to Suppl. Fig. 7b.
- **Fig. 8** is new, showing a model of the conditions under which RNF10 promotes RPS3 monoubiquitination, as well as dissociation and turnover of stalled 40S subunits.

Response to reviewers:

Reviewer #1:

Lehmann et al describe effects of RNF10-mediated RPS3 ubiquitination upon treatment with (1) translation elongation inhibitors and (2) under conditions of ribosome biogenesis perturbation. While most of the results for point (1) have previously been published (Garzia et al 2021 and Garshott et al 2021), there is some new characterization of the action of RNF10 in resolving half-mer ribosome populations under these conditions. Additionally, ribosome biogenesis perturbation (hitting both the large and small subunit) appears to impact RNF10 levels in a reciprocal fashion suggesting a coupling of these processes. Finally, the authors show that RNF10 overexpression leads to decreased levels of polysomal half-mers upon knockdown of 60S proteins.

The new findings of this paper are the links between RP levels (biogenesis) and RNF10 levels, as well as the impact of RNF10 overexpression on half-mer formation. However, RNF10-mediated ubiquitination of RPS3 has been previously characterized both in yeast and in mammals thus detracting from the novelty of the study. The new observations are generally interesting, but the depth of insight is limited and several key questions are underexplored. For instance, the half-mers formed in distinct conditions of elongation inhibition and RPL knockdown likely reflect partially resolved collisions or iRQC complexes, respectively (where the 40S subunit is either lagging or leading in the complex). The new data suggest that these two different complexes may both be processed by RNF10. How do different half-mer complexes (resulting from RPL KD or elongation inhibition) get cleared? Moreover, how significant is RNF10's endogenous role? Many of the results rely on overexpression of RNF10 and thus the significance of RNF10's endogenous role is rather unclear.

Overall, this a clearly written manuscript on an interesting topic with some strong experimental data, and some substantial overlap with previous work. There are also some new observations about RNF10 regulation though these are relatively under characterized and only discussed speculatively in the discussion.

Major comments

1. Many of the results have been shown by previous publications, including Garzia et al, 2021 and Garshott et al, 2021. While these publications are mentioned, their citation is lacking throughout, especially where figures show the same results that have been published. The contributions of published papers should be clearly acknowledged throughout. Below are results from this manuscript that have been previously published:

> Response: We would like to point out that we did mention these two studies both in our introduction and discussion section. When we initiated our study in 2019, RNF10 had not yet been described as an RPS3 ubiquitin ligase. At that time, the Inada lab had just published that the yeast E3 ligase Mag2 monoubiquitinates uS3 (RPS3) of stalled ribosomes (Sugiyama et al, 2019). By

homology search we found human RNF10 to be the closest homolog of Mag2 (see Reviewer Table 2 below, response to reviewer 4), and started to investigate its role upon ribosome stalling. While our work was still in progress, Garzia et al (2021) and Garshott et al (2021) published their findings on RNF10 as a human RPS3 ubiquitin ligase.

On the one hand, our findings are in line with several of the results obtained by Garzia et al (2021) and Garshott et al (2021), though it is important to note that we obtained our results independently, and in different cell lines: HeLa, HCT116 and RPE1 in our case, HEK293 in Garzia et al (2021) and HEK293T in Garshott et al (2021). Hence, this part of our manuscript serves as a validation of RNF10 specificity, and we now made sure to refer to these two studies more thoroughly both in the introduction and results section. In addition, we now also refer to a very recent study, where ubiquitination of RPS3 by RNF10 was confirmed *in vitro* using reconstitution of stalled ribosome complexes (Miścicka et al, 2024, PMID: 38366554).

On the other hand, there are also substantial differences between our experiments and the work published by Garzia et al (2021) and Garshott et al (2021). First, we carried out polysome profile analysis at high resolution, and thereby were able to detect ribosomal half-mer formation during translation initiation and elongation stalling. This allowed us to reveal the role of RNF10 in dissociation of stalled 40S subunits.

Second, we not only used externally applied pharmacological inhibitors for the induction of ribosomal stalling and half-mer formation, but also a cell-intrinsic approach by reducing the 60S pool through siRNAs against RPLs. Thereby, we could demonstrate that the expression level of RNF10 responds to the availability of functional 40S and 60S ribosomal subunits. Hence, our study connects RNF10 not only to translation stalling, but also to the disturbance of ribosome biogenesis.

Third, in the revised manuscript we now provide evidence for two phenotypes in RNF10 KO cells: i) Cell growth measurements showed that RNF10 KO cells have a small proliferation defect under normal conditions (~20 % reduction), and a massive proliferation defect (60-70 % reduction) when grown at low concentrations of anisomycin (new **Fig. 2f**; described in more detail below). ii) Careful quantification of polysome profiles of cells grown in presence of low dose anisomycin revealed that RNF10 KO cells have elevated amounts of ribosomal half-mers (new **Fig. 2c, d and e**; described in more detail below). This is in perfect agreement with our finding that RNF10-WT overexpression reduces the amount of half-mers (**Fig. 2b, c and e**). By revealing these phenotypes in RNF10 KO cells, our manuscript now demonstrates an essential and non-redundant role of RNF10 in dissociation of stalled 40S subunits and cellular adaptation to chronic perturbation of protein synthesis. We hope this reviewer agrees that we were able to significantly improve our manuscript with these findings, going beyond what the two previous publications showed.

- In Fig. 1A, the effect of elongation inhibitors on RPS3-Ub has been shown by Garzia et al, 2021.

> Response: While Garzia et al (2021) used three initiation inhibitors (harringtonine, hippuristanol and silvestrol) and anisomycin as an elongation inhibitor, we made use of the elongation inhibitors anisomycin (blocks PTC), cycloheximide (blocks tRNA E-site) and blasticidine (blocks tRNA P-site). Thus, our data are only in part redundant to what was shown by Garzia et al (2021).

- Similar results to Fig. 1C have been shown by Garzia et al and Garshott et al. While I appreciate that anisomycin treatment is a (slightly) different condition from published results, ubiquitination of RPS3 in ribosomal fractions has been shown and this should be acknowledged.

> Response: Garzia et al (2021) showed ribosomal fractions of FH-RNF10-expressing HEK293 cells in the absence of translation inhibitors, while we show ribosomal fractions of HeLa-RNF10-KO cells in the absence and presence of ANI (KO clones in former Fig. 1c, KO pools in new **Fig. 1c**). Garshott et al (2021) did not show polysome fractions of cells treated with elongation inhibitors such as ANI. Rather, these authors presented polysomal fractions of HEK293T cells treated with harringtonine, a translation initiation inhibitor. Their profiles were not resolved to high resolution, and the 60S peak was not distinct but overlapped with the surrounding subpolysomal fractions. As stated above, we now refer to these two studies also in the results section. Again, our findings are in line with, but go beyond these two publications.

- The findings of Fig. 1D are not new, as Garzia et al published that K214 is the residue on RPS3 that is ubiquitinated.

> Response: Sugiyama et al (2019) identified the uS3 monoubiquitination site on K212 in yeast. We then hypothesized that the corresponding RPS3 monoubiquitination site in human might be K214. We verified this in our lab by expressing RPS3-K214A in HeLa cells in July/August 2021, shortly before Garzia et al (2021) published their results in HEK293 cells. We believe that independently verifying such results, using e.g. a different cell line, is important to consolidate scientific knowledge. We now made sure to cite Garzia et al (2021) at this point in the results section.

- Garzia et al should be cited for the C225S/C228S mutant of RNF10.

> Response: We generated our RNF10 C225S/C228S mutant before their results were published in 2021. The RNF10 amino acid sequence consists of a RING finger domain with cysteines that are reported to contribute to the catalytic activity of E3 ubiquitin ligases (Garcia-Barcena et al, 2020; PMID: 32117970). Hence, it was plausible to mutate the first two cysteines of this RING domain, C225 and C228, in order to abolish the catalytic activity of RNF10. We now also cite Garzia et al (2021) in the context of the C225S/C228S mutant.

- In Fig. 2, increased Ub-RPS3 and increased free 60S subunits upon RNF10 overexpression were observed by Garzia et al, as was the lack of phenotype of RNF10 KO.

> Response: We would like to point out that we made these observations independently of Garzia et al (2021), and before their work was published. Former Fig. 2c, quantifying the increase of 60S subunits upon RNF10-WT overexpression, was now moved to the supplement (**Supplementary Fig. 2a**).

Beyond the findings of Garzia et al (2021), we additionally detected the absence of ribosomal half-mer formation in the RNF10-WT overexpressing cells upon inhibition of translation elongation (ANI treatment, **Fig. 2b, c and e**) or translation initiation (RPL7 KD, **Fig. 7a-c**). These findings then became the focus in our manuscript as they reveal a novel role of RNF10 in dissociation of stalled 40S subunits.

As described in more detail below, we could now also observe phenotypes in the RNF10 KO cells (elevated half-mer formation, new **Fig. 2c-e**; translation-sensitive proliferation defect, new **Fig. 2f**). Thus, our study represents an important step forward in understanding the function of RNF10 in mammalian cells.

- In the discussion of Fig. 6 in the text, there needs to be citation of Garzia et al. in finding that RNF10 activity responds to initiation and elongation stalls.

> Response: In the revised manuscript, we now made sure to cite both Garzia et al (2021) and Garshott et al (2021) in discussing the activity of RNF10.

2. RNF10 KO does not alter many of the phenotypes explored in this manuscript. While RNF10 KO prevents RPS3 ubiquitination upon translation elongation inhibition (Fig. 1), this does not alter half-mer formation, 40S subunit levels, or global translation more generally, as assessed by polysome profiling. Most of the strong phenotypes of translation dynamics characterized here only occur upon very strong RNF10 overexpression. This may suggest that at endogenous levels, RNF10's contribution to translation and/or ribosome dynamics is minor. To ask whether RNF10 is a major factor in the response to elongation stress or 60S biogenesis perturbation, the authors could test whether RNF10 KO impacts protein output or ribosomes levels over a longer period of stress. At the very least, the authors should discuss what the lack of phenotype in RNF10 KO cells may mean for how endogenous levels of RNF10 contribute to ribosome homeostasis.

> Response: We agree with the reviewer that demonstrating a phenotype in RNF10 KO cells would allow to establish a non-redundant function for this protein, and that chronic stress is a condition worth pursuing. To address this aspect, we first generated pools of our RNF10-KO cell lines by combining five different single RNF10-KO clones: HeLa-RNF10-KO (29-pool) with clones 29-3, 29-4, 29-6, 29-7 and 29-12; and HeLa-RNF10-KO (81-pool) with clones 81-4, 81-7, 81-10, 81-12 and 81-15. The goal was to average out potential clonal effects since phenotypes might be subtle.

With these KO pools, we explored anisomycin treatment (0.1 $\mu\text{g/ml}$ ANI) for longer periods of time and observed RPS3 monoubiquitination after 6 and 10 h of treatment in parental HeLa cells and the control KO pool, but not the RNF10 KO pools (new **Supplementary Fig. 1f**). During these experiments, we noticed that cells started to die after more than 10 h of ANI treatment at 0.1 $\mu\text{g/ml}$. Therefore, we reduced the dose of ANI to 0.02 $\mu\text{g/ml}$, at which cells can survive up to 72 hours. As suggested by the reviewer, we measured protein production under such conditions of chronic translational stress (0.02 $\mu\text{g/ml}$ ANI, 48 hours), yet did not observe a systematic difference between the KO pools and control cells by puromycin incorporation (Reviewer Fig. 1).

We additionally performed polysome profile analyses under chronic stress conditions with low dose ANI (0.02 $\mu\text{g/ml}$, 48 hours). Again, we observed that RNF10-WT overexpression prevented ribosomal half-mer formation (new **Fig. 2c**), as seen with acute translational stress (ANI 0.1 $\mu\text{g/ml}$, 2 hours, **Fig. 2b**). Systematic quantification of these polysome profiles further showed that ribosomal half-mers are elevated in the two RNF10-KO pools compared to parental HeLa cells (new **Fig. 2c, d and e**). The increase we measured is in the range of 12-19%, and is statistically significant. Thus, our new measurements revealed a functional phenotype in RNF10-KO cells, demonstrating that the endogenous levels of RNF10 promote 40S dissociation.

Since translational stress and ribosome stalling can also impair cell cycle control (Stoneley et al, 2022, PMID: 35180429), we further tested whether RNF10 contributes to cell proliferation during chronic translational stress. Proliferation of parental HeLa cells, the two RNF10-KO pools and the RNF10-WT overexpressing cells was measured in the absence and presence of low dose ANI (0.02 $\mu\text{g/ml}$) over a period of 72 hours. Under basal conditions, proliferation of the RNF10-KO pools was reduced by approximately 20%. In the presence of low dose ANI, proliferation of the RNF10-KO pools was reduced much more strongly, by approximately 60-70% (new **Fig 2f**). These effects were fully rescued in the KO cells where RNF10-WT expression was restored. Our new results thus show that RNF10 plays an essential role in sustaining cell proliferation during chronic translational stress, revealing a second physiological phenotype of RNF10-KO cells.

3. Several points made by the new data in this manuscript are underexplored.

- The correlation of RNF10 levels with 40S/60S levels is interesting, but RNF10 degradation is not explored beyond proteasome inhibition.

> Response: In **Fig. 4** (former Fig. 3), we show that RNF10 protein levels correlate with the amount of 40S subunits and that impairment of 40S biogenesis by KD of RPS leads to proteasomal degradation of RNF10. To further pursue the mechanism of RNF10 degradation, we now tested whether RNF10 ubiquitinates itself upon KD of RPS. To this end, we made use of our KO+RNF10-WT and KO+RNF10-Mut cells (lacking the E3 ubiquitin ligase activity). Since these cell lines express transgenic RNF10-WT and -Mut in the RNF10-KO background, there is no endogenous RNF10 activity. We knocked down RPS3 in these cells and simultaneously applied the proteasome inhibitor MRZ. **Reviewer Fig. 2** shows that the transgenic proteins are not reduced in their expression by RPS3 KD, indicating that proteasomal degradation of endogenous RNF10 cannot be recapitulated with the transgenic proteins. This could be due to two reasons: either the HA tag at the N-terminus of the transgenic protein precludes recognition by the ubiquitin/proteasome system, or the elevated expression levels compared to endogenous RNF10 overwhelm the ubiquitin/proteasome system responsible for degrading RNF10.

The Western blot in **Reviewer Fig. 2** also shows that MRZ treatment causes a smear above the RNF10-WT signal, and more strongly above the RNF10-Mut signal, independently of RPS3 KD. This may indicate that another, unidentified E3 ligase polyubiquitinates the RNF10-WT and RNF10-Mut proteins. Whether ZNF598 may target RNF10 for proteasomal decay is addressed in the **Reviewer Fig. 3** (comment below). Moreover, we noticed that treatment with MRZ reduced RPS3 monoubiquitination (**Reviewer Fig. 2**), indicating that proteasome inhibition generally interferes with the recognition and/or ubiquitination of stalled 40S subunits. Taken together, it turned out to be very difficult to further explore RNF10 degradation. Alternative approaches such as knock-in of RNF10 mutations will be needed to address the underlying mechanism, which is beyond the scope of our revision.

i. How rapidly is RNF10 degraded upon knockdown of 40S proteins? Does this correlate with 40S protein levels? A time course of these protein levels upon 40S protein knockdown would be useful.

> Response: We now performed time course experiments upon KD of RPS3 and RPS20, and observed that RNF10 is degraded within approximately 18 hours (**new Fig. 4d**). While RPS20 shows a decline to about 50% after 24-30 hours, the decay of total RPS3 is barely visible over the 30-hour time course, and can only be observed in the free 40S-associated fraction (**Fig. 4a**, polysomes recorded after 16 h of KD). Thus, RNF10 levels decline much faster than those of RPS3 or RPS20. Our interpretation is that mature 40S particles synthesized before disruption of 40S biogenesis remain stable, while RPS3 / RPS20 KD generates defective 40S particles that are restricted to the free 40S fraction and presumably degraded through pre-40S biogenesis control (Strunk et al., 2012, PMID: 22770215; Ameismeier et al., 2018, PMID: 29875412). Possibly, RNF10 is also degraded during this process. The rapid decline of RNF10 levels upon RPS KD indicates that RNF10 degradation is linked to the appearance of defective 40S subunits rather than to the decline of total RPS3/20 protein levels.

ii. Upon 60S protein knockdown, are increased RNF10 levels directly dependent on further stabilization? Testing whether proteasome inhibition impacts RNF10 levels +/- RPL knockdown could help parse whether RNF10 levels are impacted by increased stability or potential increased protein synthesis.

> Response: To address this point, we performed KDs of RPL7, RPL11 and RPL5 in the absence and presence of the proteasome inhibitor MRZ. In our original submission, we already showed that

RNF10 levels increase upon KD of RPLs in the absence of MRZ (now **Fig. 5 and 6**). In the presence of MRZ, we now observed a slight decrease of RNF10 protein levels (new **Suppl. Fig. 5e-g**). While the differences between - and +MRZ were statistically not significant, the pattern indicates that proteasome inhibition might interfere with the induction of RNF10 protein levels upon RPL KD. Notably, we had to restrict the duration of MRZ treatment to 8 hours since cells started to die afterwards. Western blot analysis of K48-linked ubiquitin confirmed that MRZ was active in our experiments (new **Suppl. Fig. 5f**). Interestingly, this blot also showed that MRZ prevents RPS3 monoubiquitination upon RPL KD, suggesting that proteasome inhibition interferes with the recognition and/or ubiquitination of 40S subunits stalled at the level of initiation. Since there is no increase in RNF10 levels upon MRZ treatment under control conditions (**Fig. 4e**, new **Suppl. Fig. 5f and g**), we conclude that RNF10 is normally not subject to proteasomal decay, hence it is unlikely that the increase in RNF10 levels upon RPL KD is caused by further stabilization.

iii. The mechanism of RNF10 degradation could be clarified using a diGly proteomics approach to identify ubiquitinated residues or screening a library of siRNAs targeting E3 ligases to clarify how RNF10 is targeted to the proteasome.

> Response: RNF10 was found to be ubiquitinated at several different sites within its amino acid sequence as shown in the protein modification data base PhosphoSitePlus (<https://www.phosphosite.org>). Performing serial mutagenesis to test potential lysine residues implicated in RNF10 degradation is hampered by the fact that transgenic HA-tagged RNF10 (unlike endogenous RNF10) is not subject to decay upon RPS KD (**Reviewer Fig. 2**). A DiGLY proteomics screen (see Fulzele and Bennett, 2018, PMID: 30242721) is a larger undertaking and beyond the scope of this manuscript.

To get some insight into the possible role of a candidate E3 ligase, we tested whether ZNF598 might be required for targeting RNF10 for proteasomal degradation by co-knockdown (co-KD) of ZNF598 and RPS3 in HeLa cells. However, we detected no differences in RNF10 levels after co-KD of ZNF598 and RPS3 in comparison to RPS3 KD alone (**Reviewer Fig. 3a**, compare lanes 4 and 6). We then performed sequential KD of ZNF598, followed 24 h later by KD of RPS3 for 16 h. In this set-up, we observed an approximately 1.3-fold increase of RNF10 levels by the initial KD of ZNF598 under basal conditions (second si-RNA = control, lanes 1 and 2 in **Reviewer Fig. 3b and c**), and an approximately 1.7-fold increase under conditions of RPS3 KD (lanes 3 and 4 in **Reviewer Fig. 3b and c**). However, none of these differences were statistically significant based on our three repeat experiments.

These results may suggest that ZNF598 could be involved in targeting RNF10 for proteasomal degradation upon RPS KD. However, we noticed that the outcome of this experiment strongly depends on the exact timing of the KDs (compare **Reviewer Fig. 3a with b**). Moreover, the effect of ZNF598 KD could be indirect. Given that these data are not conclusive at this point, we prefer not to show them in the manuscript. We do show the increase of RNF10 expression upon ZNF598 KD in RPE1 cells (new **Suppl. Fig. 1c and d**), which might point in a similar direction. Resolving the possible role of ZNF598 in RNF10 degradation will require recombinant expression of these proteins. We extensively tried to produce recombinant human RNF10 in *E.coli*, but were not successful despite various attempts (see response to the second reviewer's second comment, below). An unbiased screen using a library of siRNAs or gRNAs targeting E3 ligases is an entire new project that cannot be addressed within this revision.

• It is interesting that RNF10 overexpression can decrease half-mers both upon elongation inhibition and RPL knockdown. How RNF10 clears these distinct complexes is unclear.

> Response: This is no doubt a challenging question. To obtain further insight into the mechanisms by which RNF10 promotes 40S dissociation, we first tested whether ubiquitination of RPS3 at K214 is responsible for the dissociation of ribosomal half-mers. To this end, we performed a prolonged KD of endogenous RPS3 using an siRNA (S292) that targets the 3'UTR of the endogenous, but not the exogenous transcript. HeLa-FLAG-RPS3-WT and -K214A cells were transfected with this siRNA twice over a period of 72 hours to allow for replacement of endogenous RPS3 with the exogenous copy. Cells were then exposed to either medium dose (0.1 $\mu\text{g/ml}$, 2 h) or low dose ANI (0.02 $\mu\text{g/ml}$, 48 h) before polysome profiles were recorded. The first observation we made in this set-up is that the HeLa-FLAG-RPS3-WT cells disassembled stalled ribosomes more efficiently since half-mer formation was not observed upon ANI treatment (new Fig. 3b). In contrast, half-mers were visible in the HeLa-FLAG-RPS3-K214A cells (Fig. 3b, quantified in new Fig. 3c). We further noticed that the free 40S peak was elevated upon ANI treatment in RPS3-K214A cells compared to RPS3-WT cells (also quantified in Fig. 3c). Thus, our new data provide evidence that dissociation of ribosomal half-mers and clearance of stalled 40S subunits depends on ubiquitination of RPS3 at K214.

Second, we pursued the question whether RPS3 might be polyubiquitinated. In the context of yeast non-functional rRNA decay (NRD), monoubiquitination of RPS3 at K212 by Mag2 (human RNF10) is followed by polyubiquitination through the E3 ubiquitin ligase Hel2 (human ZNF598) upon ribosome collisions, or through the E3 ubiquitin ligase Fap1 upon individually stalled ribosomes, both of which results in degradation of the non-functional 18S rRNA (Li et al, 2022, PMID: 36113412). The human homolog of Fap1 is the E3 ligase and transcriptional repressor NFX1 (PMID: 36113412), yet it is currently unknown whether NFX1 plays a similar role in tagging non-functional ribosomes in mammals. We therefore tested whether polyubiquitinated RPS3 can be detected under different conditions of ribosome perturbation and stalling. In the Western blot of Reviewer Fig. 4a, several bands above RPS3 and its monoubiquitinated form can be seen, including a smear close to the stacking gel. However, none of these higher bands was dependent on RNF10. Likewise, we were also unable to detect polyubiquitinated RPS3 dependent on ZNF598 (Fig. 1a, Suppl. Fig. 1h, Reviewer Fig. 3a).

We then performed KD of NFX1 with two different siRNAs, followed by treatment of cells with the translation initiation inhibitor Rocaglamide (Roca, an eIF4A antagonist) or the elongation inhibitor ANI. Again, bands migrating above Ub₁-RPS3 were monitored, yet no NFX1-dependent changes could be observed under either condition (**Reviewer Fig. 4b and c**). Thus, our experiments did not provide any evidence for stalling-induced, RNF10-, NFX1-, or ZNF598-dependent RPS3 polyubiquitination. We interpret the bands migrating above Ub₁-RPS3 as mostly derived from cross-reactivity of the RPS3 antibody with other proteins.

i. Upon elongation inhibition, Ub-RS3 is associated with polysomes but not with the free protein fraction (Fig. 1c); upon RPL7 knockdown, the converse is true (Fig. 4a). Could the authors comment on how this may relate to the mechanism of RNF10-mediated ubiquitination in these different conditions?

> Response: This comment now pertains to **Fig. 1c and 5a**. We believe that the difference in Ub₁-RPS3 distribution is a direct consequence of ribosome elongation kinetics: With the elongation inhibitor ANI, which at the moderate dose of 0.1 µg/ml induces ribosome stalling of a few, but not all, elongating ribosomes, numerous ribosomal collisions occur along the entire ORF. Since ribosomes cannot further elongate, the mRNAs remain "stuck" in the polysomal fractions, and monoubiquitinated RNF10 will be visible throughout the polysomal part of the gradient.

In the case of RPL7 knockdown, stalling occurs primarily at the start codon by 40S subunits "waiting" for 60S subunits to join. Ribosomes that had previously initiated on these mRNAs (using intact 60S subunits that remain from the period before RPL7 KD) will elongate normally and "run off" the mRNA within a short time (few minutes given an elongation rate of approximately 5-6 codons per second (Ingolia et al., 2011, PMID: 22056041)). As a consequence, these mRNAs will be located primarily in the 40S fraction (48S to be precise, though this is beyond the resolution of sucrose density gradients), and a smaller proportion still containing elongating ribosomes will be located in the half-mer fractions of smaller polysomes (1.5-mer, 2.5-mer, 3.5-mer). We now show a direct comparison of polysome fractionation blots from cells subjected to ANI and RPL7 KD cells, where the difference in Ub₁-RPS3 distribution is very well visible (new **Suppl. Fig. 5d**). In this blot, one can observe the Ub₁-RPS3 distribution upon RPL7 KD as predicted: a major proportion is located in the 40S fraction (#2), and a small proportion in smaller polysomes (visible as faint bands in fractions 5-7). In contrast, the Ub₁-RPS3 signal goes far more into the polysome fraction in ANI-treated cells. This difference is now also visualized in a quantification of the Ub₁-RPS3 distribution (new **Fig. 5g**).

To obtain further evidence for this kinetic model, we visualized the distribution of EFD1 across the polysomal fractions. EFD1 is a collision sensor that binds to stalled collided di-ribosome complexes (Sinha et al., 2020, PMID: 32744497). Upon ANI treatment, EFD1 accumulated in polysomal fractions with a distribution similar to Ub₁-RPS3 (**Suppl. Fig. 5d**), confirming the arrest of elongating ribosomes. In contrast, EFD1 remains predominantly associated with sub-polysomal fractions upon

RPL7 KD, resembling the distribution in control cells (**Suppl. Fig. 5d**). This confirms that RPL7 KD primarily leads to stalling of the initiating 40S (48S) subunit, without causing ribosomal (80S-80S) collisions. With these results, we provide additional evidence that RNF10 does not only respond to the stalling and collision of elongating ribosomes, but also to initiation stalling at the level of 48S pre-initiation complexes.

Under both ANI and RPL7 KD conditions, we also observe Ub₁-RPS3 in the first fraction comprising free proteins. For one, this is due to "bleeding" of the 40S peak into fraction #1. For another, Ub₁-RPS3 might also be liberated from stalled ribosomes following 40S dissociation and turnover. It is very unlikely that RNF10 monoubiquitinates free RPS3 protein since RNF10 activation occurs only upon initiation or elongation stalling.

ii. Additionally, upon 60S biogenesis perturbation, it is not completely clear whether RNF10 is ubiquitinating RPS3 associated with free 40S subunits or 40S subunits stalled on mRNAs. Ub-RPS3 is not associated with polysomal half-mers upon RPL7 KD (Fig. 4a), yet RNF10 overexpression clears these polysomal half-mers (Fig. 6c) – how do the authors parse this discrepancy? The authors should test whether RPL7 KD with RNF10 overexpression induces Ub-RPS3 associated with polysomal half-mers.

> Response: This comment now pertains to **Fig. 5a and 7a**. As outlined in our response to the previous question, we observe that small amounts of Ub₁-RPS3 are associated with polysomal fractions containing the half-mers (fractions 5-7, new **Suppl. Fig. 5d**, quantified in new **Fig. 5g**). Hence, our results are compatible with the model that RNF10 ubiquitinates RPS3 of 40S (48S) subunits stalled at the start codon, and promotes their dissociation when RNF10 is overexpressed.

We followed the reviewer's suggestion to test whether RPL7 KD in RNF10-WT overexpressing cells induces RPS3 monoubiquitination in association with polysomal half-mers. The result is shown in **Reviewer Fig. 5**. Substantial amounts of Ub₁-RPS3 can be observed in polysomal fractions when RNF10-WT is overexpressed (fractions 5-9), compatible with our model.

4. Some controls are lacking. For example, in Fig. 3, the accounting of protein levels after siRNA KD should be more careful. RPS3 protein levels from whole cell lysate shown be shown for Fig. 3A/B. Indeed, KD of RPS3 does not appear to have a significant impact on RPS3 levels in Fig. 3C. It is understandable that KD of a ribosomal protein would not yield cells with a very strong knockdown, but it warrants comment. Quantification of RP levels upon KD would be helpful to understand the degree of knockdown.

> Response: This comment now pertains to **Fig. 4** and **Suppl. Fig 4**. Former Fig. 3b is now shown in **Suppl. Fig. 4a**. We now included the Western blot for RPS3 in this figure, and added a quantification of RPS3 and Ub₁-RPS3 levels to this panel. RPS3 KD causes only a weak reduction of the total RPS3 signal, down to about 60-80% (this is 24 hours after siRNA transfection). However, one can see a much stronger reduction of RPS3 in the subpolysomal fractions (#1-3) following sucrose density gradient centrifugation (**Fig. 4a**). These fractions mostly comprise the newly synthesized 40S subunits (lacking RPS3), while all ribosomes engaged in polysomes contain the intact 40S subunits from the period before siRNA transfection. Since ribosomes are very stable, the turnover of ribosomal proteins is slow, which is quite impressively visible in the Western blot from polysome fractionation. We now explain this aspect better in the text.

5. As a final point, some of the references selected for background material were surprising and not accurate. For example, the reference chosen for NGD (15) refers to work in bacteria rather than to the abundant work on this topic in eukaryotes.

> Response: We agree with the reviewer and replaced citation 15 with a more general review covering NGD in eukaryotes: Kyle T Powers, Jenn-Yeu Alvin Szeto and Christiane Schaffitzel: New insights into no-go, non-stop and nonsense-mediated mRNA decay complexes, 2020, PMID: 32688260.

Other comments

1. In Fig. S1B, could the authors comment on why RNF10 levels decrease upon ZNF598 KD?

> Response: This comment now pertains to **Suppl. Fig. 1c**. The Western blot we showed here previously was not representative due to loading issues. After going through all our repeats, we became aware that RNF10 increases upon ZNF598 KD in RPE1 cells. Therefore, a new blot is now shown in **Suppl. Fig. 1c**, and the quantification of RNF10 levels in RPE1 cells was added to this figure (new **Suppl. Fig. 1d**). The increase in RNF10 is approximately 1.5-fold upon ZNF598 KD in RPE1 cells.

In HeLa cells, KD of ZNF598 does not alter RNF10 expression under basal conditions (**Fig. 1a**, quantification of RNF10 shown in new **Suppl. Fig. 1a**). The effect of ZNF598 KD on RNF10 levels under conditions of RPS3 KD is addressed above in Reviewer Fig. 3.

2. In Fig. S2C, RNF10 overexpression does not appear to increase free 60S subunits as it does in other figures. Could the authors address this inconsistency?

> Response: This comment now pertains to **Suppl. Fig. 2d**. The inconsistency is a technical issue of the detector of the polysome fractionator we use, showing a high sensitivity towards temperature changes, which influence the baseline of the graph. We performed the experiment again and made sure that the temperature of the detector was adjusted to the temperature of the samples. With prior temperature adjustment of the detector, we observed an increase of free 60S subunits in RNF10-WT overexpressing cells (new polysome curves in **Suppl. Fig. 2d**), as in the other experiments.

3. KD of some small subunit proteins appears to impact levels of other small subunit proteins (see Fig. 3C,E) and this is also true for large subunit proteins (see Fig. 4D,F). This likely points to disruptions in small and large subunit stability and is an interesting point that warrants comment.

> Response: This comment now pertains to **Fig. 4** and **5**, respectively. As correctly noted by the reviewer, the expression of some ribosomal proteins seems to be affected strongly by the KD of others. This is especially true for RPS20, whose expression goes down upon KD of RPS3 and RPS6 (**Fig. 4b**), and for RPL11, whose expression goes down after KD of RPL5 and RPL7 (**Fig. 5f**). This likely points to a higher turnover of these ribosomal proteins compared to others, and their sensitivity to disruption of 40S and 60S ribosome biogenesis. In order to keep our story focused, we prefer not to comment on the intricate interdependencies and exchange rates of ribosomal proteins.

4. In Fig. S3D, could the authors speculate as to why siRPS6 and siRPS20 cause differential effects on polysome profiles?

> Response: This comment now pertains to **Suppl. Fig. 4f**. This is the same issue as described above, where the baseline at the beginning of the profiles depends strongly on temperature differences. We now repeated the experiment with prior temperature adjustment of the detector, and could improve the reproducibility of the polysome profiles. KD of RPS6 and RPS20 have very similar effects on the profiles, showing a decrease of free 40S subunits and a marked increase of 60S subunits, without major differences in the profiles (new **Suppl. Fig. 4g**).

5. In Fig. S3E, the implication is that RIO2 KD is causing decreased Ub-RPS3 via decreased RNF10 levels. An alternative explanation could be that overall RPS3 levels are decreased upon RIO2 KD, leading to decreased Ub-RPS3. Quantification of RPS3 and Ub-RPS3 levels in this experiment could help clarify this point.

> Response: Since the effect of RIO2 KD on RNF10 expression is rather mild (compared to the strong effect of RPS KD), we now decided to take out the RIO2 data from our revised manuscript. Thereby, we can streamline our story and make space for the new data.

6. Fig. 5 is not very relevant to the main conclusions of the paper and should be moved to the supplement.

> Response: This comment now pertains to **Fig. 6**. We would like to keep parts of this figure in our main manuscript since p53 activation is known to occur as a direct response to the disturbance of ribosome biogenesis (see e.g. Lindström et al, 2022, PMID: 35444234; Pelava et al, 2016, PMID: 27528756; Golomb et al, 2014, PMID: 24747423; Liu et al, 2016, PMID: 28741571; Tiu and Barna et al, 2021, PMID: 34242585). Therefore, we believe that it is important to test whether changes in RNF10 levels are dependent on p53 activity. To streamline our manuscript, we now show only panels a and b in the main part. Panel c together with a new quantification of RNF10 levels is now shown in **Suppl. Fig. 6a and b**. Panel d of former Fig. 5 was omitted since elevated expression of p53 upon RPL7 KD is not related to the gist of our manuscript. We also added a reference (Lindström et al, 2022, PMID: 35444234) to strengthen the justification for testing the involvement of p53.

• In Fig. 5D, can the authors comment on why ZNF598 levels are decreased upon treatment with any siRNA other than the control siRNA?

> Response: We repeated KDs of RPL5, RPL7 and RPL11 in HCT116 cells and could not observe a consistent decrease of ZNF598 levels (**Reviewer Fig. 6**). Since we decided to remove former Fig. 5d anyway (see response to the comment above), this point will no longer be an issue.

Reviewer #2:

Review for the manuscript NCOMMS-22-43495 by Dr. Stoecklin and co-authors entitled "The E3 ubiquitin ligase RNF10 promotes dissociation of stalled ribosomes and responds to ribosomal subunit imbalance"

It has been demonstrated that the ubiquitination of uS3 is essential for the quality control of 18S nonfunctional rRNA decay (NRD) that induces the degradation of the nonfunctional ribosome with the mutation in the decoding center. In yeast, 18S NRD depends on the polyubiquitination of uS3 by a set of ubiquitination enzymes. Mag2 is essential for 18S NRD, and responsible for the monoubiquitination followed by the polyubiquitination. Fap1 is the main E3 ligase for 18S NRD and Hel2 partially contributes to the degradation of the nonfunctional ribosome. Although the 18S NRD in higher eukaryotes is largely unknown, however, the involvement of RNF10, a human homolog of Mag2 in the degradation of the 40S subunit has been suggested. RNF10 has an activity for the monoubiquitination of uS3 at K214, the compatible residue of the yeast uS3 K212. However, it is unknown whether the monoubiquitination is followed by polyubiquitination with ZNF598 or other E3 ligases in higher eukaryotes.

In this study, the authors demonstrated that translation elongation stalling induces mono-ubiquitination of RPS3 at K214 by RNF10. RNF10 overproduction reduces the halfmers formation induced by anisomycin treatment and induced by RPL KD condition. The proposal is potentially interesting. However, the mechanism by which RNF10 reduces the halfmers is not addressed. Although RNF10 overproduction reduces the halfmer formation induced by anisomycin treatment, the RNF10-KO phenotype is not evident. Moreover, it is unclear whether the reduction of the halfmers is due to the dissociation of 40S by RNF10 simply because the dissociation of the 40S subunit from halfmers is not experimentally demonstrated. It is no experimental analysis to examine how the RNF10 level is reduced in a proteasome-dependent manner in response to the KD of RPS proteins, and how the RNF10 level is increased in response to the KD of RPL proteins. The relation between proteasomal degradation of RNF10 upon RPS-KD and upregulation of RNF10 upon RPS-KD is unaddressed. There is no mechanistic insight for the regulation but transcription factor p53 does not control RNF10 expression. Overall, the experiments that support the proposal are largely missing, more experimental evidence must be done to support the activity of RNF10 proposed in this study. It's reasonable to revise the overstatement in the discussion based on the experimental results shown in this manuscript.

Major points

1. The authors proposed that the mono-ubiquitination RPS3 K214 by RNF10 upon translation elongation stalling is involved in RNF10-mediated rescue and suppression of ribosomal halfmer formation. This assumption should be directly investigated by the experiments using RPS3-K214R mutant cells. The authors demonstrated that RNF10 overproduction reduces the halfmer formation induced by anisomycin treatment (Figure 2b) and induced by RPL KD condition (Figure 6c). The phenotype of the RPS3-K214R mutant cells will strongly support that the ubiquitination of uS3 is required for the RNF10-mediated reduction of the halfmer.

> Response: This comment now refers to **Fig. 2b** and **Fig 7a**, respectively. To test whether ubiquitination of RPS3 at K214 is responsible for the dissociation of ribosomal half-mers, we ran polysome profiles of HeLa cells stably expressing FLAG-RPS3-WT and RPS3-K214A, in the absence and presence of ANI. With this set-up, no differences in half-mer formation were detectable between FLAG-RPS3-WT and Flag-RPS3-K214A cells (data not shown). One possible explanation is that endogenous RPS3 is still expressed in these cells and may occlude effects of the K214 mutation.

Therefore, we performed a prolonged KD of endogenous RPS3 using an siRNA (S292) that targets the 3'UTR of the endogenous, but not the exogenous transcript. HeLa-FLAG-RPS3-WT and -K214A cells were transfected with this siRNA twice over a period of 72 hours to allow for better replacement of endogenous RPS3 with the exogenous copy. Cells were then exposed to either a medium dose (0.1 µg/ml, 2 h) or a low dose of ANI (0.02 µg/ml, 48 h) before polysome profiles were recorded. The first thing we noticed in this set-up is that the HeLa-FLAG-RPS3-WT cells disassemble stalled ribosomes more efficiently since half-mer formation was not observed upon ANI treatment (new **Fig. 3b**). In contrast, half-mers were visible in the HeLa-FLAG-RPS3-K214A cells (**Fig. 3b**, quantified in new **Fig. 3c**). We further noticed that the 40S peak was smaller in FLAG-RPS3-WT cells compared

to FLAG-RPS3-K214A cells (also quantified in **Fig. 3c**). Both, the absence of half-mers and the reduced 40S peak in FLAG-RPS3-WT cells resemble the polysome profiles of cells overexpressing RNF10-WT (**Fig. 2b**). Thus, our new data provide evidence that dissociation of ribosomal half-mers and clearance of stalled 40S subunits depends on ubiquitination of RPS3 at K214.

2. It is unclear whether the reduction of the halfmers is due to the dissociation of 40S by RNF10 simply because the dissociation of the 40S subunit from halfmers is not experimentally demonstrated. In the related RQC filed, *in vitro* reconstitution of ZNF598-mediated ribosome ubiquitination and subunit dissociation by the RQT complex have been reported (PMID: 32579943 PMID: 36302773). I highly recommend, to conclude RNF10 promotes the dissociation of stalled ribosomes shown in the title, the RNF10-mediated ubiquitination of the halfmers and subunit dissociation should be demonstrated by the *in vitro* reconstitution reaction.

> Response: We agree with the reviewer that an *in vitro* reconstitution system could provide further evidence for the role of RNF10 in 40S dissociation upon ribosome stalling. We have therefore tried to recombinantly express human RNF10 in bacteria, also with kind support from the Sinning group (BZH Heidelberg), which has ample experience in recombinant protein expression. However, we were unable to purify the protein because recombinantly expressed RNF10 was unstable: it quickly aggregated and degraded to smaller fragments (data not shown). We tested several different expression systems, tags, buffers, zinc concentrations, alternative codons, and also shortened the protein at the N- and C-termini, hoping to obtain stable recombinant RNF10. Unfortunately, all these approaches failed, and we had to give up the expression of recombinant RNF10 for the time being. Setting up an *in vitro* reconstitution system will need alternative attempts such as expressing the protein in insect cells, which is currently not established in our lab.

During elongation stalling, half-mers form when the 60S subunit dissociates from the leading ribosome of a collided disome through the activity of the RQT complex (Best et al, 2022, PMID: 36801861). Basically, there are two possibilities how RNF10 could suppress half-mer formation: Either, RNF10 inhibits RQT complex-mediated dissociation of 60S from the leading stalled ribosome, or RNF10 promotes dissociation of the remaining 40S subunit.

To test whether the first hypothesis might be true, we made use of a FACS-based RQC reporter system (Juszkiewicz & Hegde, 2017, PMID: 28065601). The result from this experiment shows that while KD of ZNF589 specifically reduces RQC on the poly-lysine (K^{AAA})₂₀-containing reporter mRNA, KD of RNF10 has no impact on RQC efficiency (new **Suppl. Fig. 2e**). This result is consistent with similar findings by Garcia et al. (Garzia et al, 2021, PMID: 34348161). Moreover, our manuscript shows that RNF10 promotes loss of half-mers upon initiation stalling (RPL KD, **Fig. 7**), which does not involve the RQT complex but features an "idle" 40S (48S) subunit at the start codon. In the revised manuscript, we reinforced this part by a quantification of ribosomal half-mers upon RPL7 KD (new **Fig. 7b**). Taking into account both lines of evidence, our data strongly support the second hypothesis, i.e. that RNF10 promotes dissociation of stalled 40S subunits. This interpretation is also compatible with the strong link between RNF10 and 40S subunit turnover (Garzia et al, 2021, PMID: 34348161; Garshott et al, 2021, PMID: 34469731). Along these lines, we now discuss the interpretation of our data more thoroughly in the text.

In our revised manuscript, we further provide new evidence that the KO of RNF10 increases the amount of ribosomal half-mers observed upon elongation stalling (new **Fig. 2c-e**), that the RPS3-K214A mutation prevents 40S dissociation under these conditions (new **Fig. 3b and c**), and that RNF10-KO cells have a pronounced proliferation defect when grown at low concentrations of ANI (new **Fig. 2f**). By revealing these phenotypes in RNF10-KO cells, our manuscript now demonstrates an essential and non-redundant role of RNF10, and thereby adds important aspects to the current understanding of the cellular response to ribosome stalling and the perturbation of protein synthesis.

3. In the previous report (PMID: 343481610), ZNF598-KO reduces the level of mono-ubiquitination of uS3 with the anisomycin treatment. To make clear the difference, the levels of monoubiquitinated uS3 in Hela-WT and ZNF598-KO cells with the concentration of anisomycin should be compared.

> Response: Garzia et al (2021, PMID: 34348161) showed that Ub₁-RPS3 levels are decreased in ZNF598-KO and absent in RNF10-KO HEK293 cells upon treatment with the translation elongation inhibitor ANI. In contrast, Ub₁-RPS3 levels were unaffected in ZNF598-KO cells but absent in RNF10-KO cells upon treatment with the translation initiation inhibitors Harringtonine and Hippuristanol.

In **Fig. 1a**, we show that Ub₁-RPS3 levels are not reduced upon ZNF598 KD in HeLa cells. To improve the KD of ZNF598, we now performed a double KD of ZNF598 over a period of 2 days, and observed the same: no reduction in Ub₁-RPS3 levels in the ZNF598 KD HeLa cells (**Suppl. Fig. 1i**). At this point, we can only speculate that this discrepancy might be due to differences between HEK293 and HeLa cells. Alternatively, the KO of ZNF598 (in contrast to acute KD) might lead to adaptation, e.g. altered expression levels of RNF10. In fact, in Figure 5 of Garzia et al (2021, PMID: 34348161), there is some evidence for this. We would like to emphasize that our observation is concurrent with the central findings of Garzia et al (2021) and Garshott et al (2021), i.e. that RNF10 is the main E3 ligase responsible for RPS3 monoubiquitination.

Reviewer #3:

Lehmann et al. provide evidence for a feedback loop between 40S ribosomal subunit levels and RNF10. Translation and RNF10 are disrupted using multiple orthogonal techniques, including drugs, knockdown, and overexpression. Ribosomal state is measured by monitoring ubiquitination and translation state via sucrose gradients. The authors speculate on the mechanism by which RNF10 acts upon the small ribosomal subunit but do not offer direct evidence. This study complements previous work showing that proteostasis stressors regulate 40S levels (PMID: 34469731). This is a valuable systems study of how cells maintain ribosome levels in dynamic and stressful conditions. It suggests a translation-sensitive feedback loop that may be orthogonal to better understood mechanisms such as those involving ZNF598. I recommend publication after minor changes described detailed below.

Line 246: The authors mention half-mer formation and reference figs 4a and 6a. Yet these figures panels do not clearly highlight or quantify (better) half-mers as is done in 2b or 6c. Did the authors mean to reference 6c? Figures should be updated to highlight half-mer levels.

> Response: This comment now refers to **Fig. 5 and 7**, respectively. We agree that the half-mers are not well visible in **Fig. 5a** due to overlap with the si-control curve. The same condition (labeled HeLa) is also contained in **Fig. 7a**, where the half-mers are very well visible. As suggested, we now also refer to **Fig. 7a**.

Moreover, we now provide careful quantification of half-mers with help of the QuAPPPro software developed by Schiller et al in our department (Schiller, C., Reitter, S., Lehmann, J. A., Fenzl K. & Schott, J. QuAPPPro: An R/shiny app for Quantification and Alignment of Polysome Profiles. *bioRxiv* (2024). <https://doi.org/10.1101/2024.05.02.592260>). This quantification allowed us to uncover a translational phenotype in RNF10-KO cells, where we observe a slight, but statistically significant, increase in ribosomal half-mers under conditions of prolonged exposure to low dose ANI (new **Fig. 2c-e**). Likewise, quantification of half-mers in FLAG-RPS3-WT versus FLAG-RPS3-K214A cells now provides evidence that monoubiquitination of RPS3 promotes dissociation of stalled 40S subunits (new **Fig. 3b and c**). Thus, half-mer quantification allowed us to uncover important phenotypes in RNF10-KO and mutant RPS3 cells, considerably strengthening our manuscript.

Fig 1c: Any comment on why there are increased levels of free RPS3 when RNF10 is knocked out? Is RNF10 leading to ubiquitination and degradation of unassembled RPS3?

> Response: The reviewer correctly noticed the increase of RPS3 in the first polysome fraction of the two RNF10 KO clones in former Fig. 1c. To test whether this effect is reproducible, we made fresh Western blots with the same lysates, and observed much smaller differences of RPS3 in the first fraction (**Reviewer Fig. 7a**).

Moreover, we changed the experiment in **Fig. 1c** by now using two pools of five different RNF10-KO clones each, instead of two single RNF10-KO clones, with the aim to avoid clonal effects (see also response to Reviewer 1). Western blot analysis of the polysome fractions showed very similar levels of free RPS3 in parental HeLa cells and in the two RNF10 KO pools (new **Fig. 1c**). We also quantified the RPS3 levels from the new Fig. 1c, the results are shown below in **Reviewer Fig. 7b**. Taken together, RNF10 KO does not appear to affect the polysomal distribution of RPS3, and we believe that the difference in the initial blots was the result of variations in protein extraction from fractions or unequal loading.

Fig2a: mutant RNF10 is destabilized vs WT. Please address in text.

> Response: The efficiency of lentiviral transduction can vary between different experiments and samples, which results in differences in expression levels of the transduced gene. We repeated this

experiment and observed no differences between RNF10-WT and RNF10-Mut protein levels. We now replaced this Western blot and show a new experiment in **Fig. 2a**.

An interesting question is whether RNF10 may ubiquitinate itself upon KD of RPSs since KD of RPS3, RPS6 and RPS20 leads to proteasomal degradation of RNF10 (**Fig. 4e**). To this end, we made use of our transduced HeLa KO+RNF10-WT and KO+RNF10-Mut cells (lacking the E3 ubiquitin ligase activity). These cell lines express transgenic RNF10-WT/-Mut in the RNF10-KO background and thus do not have any endogenous RNF10 activity. We knocked down RPS3 in these cells and simultaneously applied the proteasome inhibitor MRZ. **Reviewer Fig. 2** (above) shows that transgenic RNF10 is not reduced in its expression by RPS3 KD, indicating that proteasomal degradation of endogenous RNF10 cannot be recapitulated with the transgenic protein. This could be due to two reasons: either the HA tag at the N-terminus of the transgenic protein precludes recognition by the ubiquitin/proteasome system, or the elevated expression levels compared to endogenous RNF10 overwhelm the ubiquitin/proteasome system responsible for degrading RNF10. **Reviewer Fig. 2** also shows that treatment with MRZ reduced RPS3 monoubiquitination, indicating that proteasome inhibition generally interferes with the recognition and/or ubiquitination of stalled 40S subunits.

Discussion section: A cartoon would be useful for summarizing the findings of this work and communicating potential molecular mechanisms.

> Response: We now present a model that depicts the two conditions under which we observe RNF10-mediated dissociation of stalled 40S subunits (new **Fig. 8a and b**), as well as a cartoon of how RNF10 helps to maintain the stoichiometric balance between 40S and 60S subunits (new **Fig. 8c**).

Reviewer #4:

The E3 ubiquitin ligase RNF10 promotes dissociation of stalled ribosomes and responds to ribosomal subunit imbalance submitted by Lehmann et al. focuses on the role of RNF10 in RPS3 monoubiquitylation and its relation to ribosomal subunit stoichiometry regulation.

This manuscript presents results from extensive cellular studies that in addition provided basis for analysis of mRNA levels and polysome profiling. These experiments uncovered interesting relationship between levels of 40S/60S subunits and RNF10 and show that RNF10 is responsible for clearance of 40S subunit via RPS3 ubiquitylation at K214.

Manuscript is overall well written and nicely organised with a good flow. Several suggestions were made below to help improve the logic of the story and overall understanding of the text. Considering that authors address these comments, I am happy to recommend this manuscript for publication.

General comments:

Authors in the introduction describe RNF10 as protein characterised in the nervous system, however, in the subsequent result sections use cell lines derived from cervical, retinal, and colorectal cell lines. Additionally, considering information available in the Human Atlas Database (<https://www.proteinatlas.org/ENSG00000022840-RNF10/tissue>) and the wide expression pattern, it would be helpful to understand the logic behind the cell line choices made. If there was preceding cell line screening, it might be helpful to disclose this information.

> Response: The Human Atlas Database indicates that RNF10 is expressed in almost all tissues at a medium level; only smooth muscle, skeletal muscle and adipose tissue seem not to express RNF10. The work we present here is primarily done in the epithelial human cervix carcinoma cell line HeLa, with some additional work in the epithelial human colorectal cancer cell line HCT116 and the non-cancerous retinal epithelial cell line RPE1. All of these cell lines express levels of RNF10 that are easily detectable by Western blot analysis. As we believe that RNF10 has a general role in the response to translation initiation and elongation stalling, we conducted most of our biochemical work in HeLa cells since genetic manipulations are efficient (CRISPS/Cas9-mediated KO, siRNA-mediated KD, lentiviral transduction) in this cell model. Experiments done in RPE1 and HCT116 cells allowed us generalize our findings.

Considering that the authors focus on the monoubiquitylation of RPS3 it would be helpful to include a representative western blot for each cell line showing the whole membrane in the supplementary materials. Highly active E3 ligases tend to form polyUb chains that are usually resolved between the higher MW regions and the stacker. If there are any findings, it would be also interesting to bring this into the context with the two-step mechanism proposed for Mag2 polyubiquitylation.

> Response: We now show entire membranes stained with the RPS3 antibody in **Reviewer Fig. 4** (above). Monoubiquitination of RPS3 is the only clear pattern we can discern, this band becomes distinctly visible after treatment with ANI and after RPL7 KD (**Reviewer Fig. 4a**). Several additional bands are visible with this antibody (at approximately 32 kDa, 55 kDa, 60 kDa, 68 kDa and 140 kDa), which we believe are due to cross-reactivity of the antibody. Moreover, on some membranes one can see a slight smear at the lower end of the stacking gel, which may, or may not, represent polyubiquitinated RPS3. Since this smear does not increase under conditions where RPS3 monoubiquitination is induced, it does not seem to be related to the stalling events we are studying.

To further assess the possibility of RPS3 polyubiquitination, we knocked down NFX1, the human homolog of the yeast E3 ubiquitin ligase Fap1, which was shown to polyubiquitinate RPS3 following monoubiquitination at K212 by Mag2 during 18S NRD in yeast (Li et al, 2022, PMID: 36113412). **Reviewer Fig. 4b and c** show that the smear at the top of the RPS3 blot was not altered upon NFX1 KD. Hence, we do not have evidence for RPS3 polyubiquitination in our system, nor for the involvement of NFX1, and hence did not further pursue this question. Since the upper parts of the RPS3 blots do not reveal any new or conclusive information, we prefer to not include them into our manuscript.

I am wondering if it is possible for the reader to compare individual polysome profiling figures. Are the ranges on Y-axes of relative units for the absorbance at 254 nm in the same range? Was the data normalised in any way?

> Response: For alignment of the polysome profiles, we used the quantification tool QuAPPPro, a bioinformatics application developed in our department (Schiller, C., Reitter, S., Lehmann, J. A., Fenzl K. & Schott, J. QuAPPPro: An R/shiny app for Quantification and Alignment of Polysome Profiles. *bioRxiv* (2024). <https://doi.org/10.1101/2024.05.02.592260>). This application allows to normalize the height and length of aligned polysome profiles, to normalize graphs using the total area under the curve, and to calculate the area under any part of the curve. Quantifications in our manuscript are always based on ratios between different areas under the curve, thereby ensuring relative normalization.

Based on QuAPPPro, our new quantification of ribosomal half-mers is illustrated in a cartoon in the new **Fig. 2d**. Using this quantification approach, we were able to uncover a translational phenotype in RNF10-KO cells, where we observe a slight, but statistically significant, increase in ribosomal half-mers under conditions of prolonged exposure to low dose ANI (new **Fig. 2c and e**). Likewise, quantification of half-mers in RPS3-WT versus RPS3-K214A cells now provides evidence that monoubiquitination of RPS3 promotes dissociation of stalled 40S subunits (new **Fig. 3b and c**). Thus, half-mer quantification allowed us to uncover important phenotypes in RNF10-KO and mutant RPS3 cells, considerably strengthening our manuscript.

Specific comments:

Authors mentioned structural interface between small subunits (ref. 18). Considering this publication and additional available structural data, two question arise for me:

1. It appears that S3 proteins and their K212 residues are in a very close proximity at the interface. Do authors think that this proximity plays any role in the regulation of Ub attachment?

> Response: It is indeed interesting to note that the C-terminal portion of RPS3 is in close proximity to RACK1, an essential component of RQC (see e.g. Sundaramoorthy et al, 2017, PMID: 28132843; Sinha et al, 2020, PMID: 32744497), as well as to the interface of collided ribosomes. Since we observe Ub₁- RPS3 not only under conditions of elongation stalling (ANI, CHX, BLA, **Fig. 1a**), but also upon 40S (48S) initiation stalling (KD of RPL5, RPL7, RPL11, **Fig. 5**), the interface of collided ribosomes does not seem to be essential for RPS3 monoubiquitination. The entirely different architecture of these two conditions is now also illustrated in new **Fig. 8a and b**. We now address the position of RPS3 in the discussion, and propose that its localization in the 40S mRNA entry

channel places it in an ideal position to monitor the presence of mRNA as a feature that is common to both types of stalled 40S subunits.

2. Considering that authors probed RPS3, RPS6, RPS19, and RPS20 and their effect on RNF10 levels, I wonder how the choice of these particular structural proteins was made. Interestingly, upon mapping these on the available structures, RPS20 seems to contact RPS3. Additionally, similar question arises for the large subunit RPL protein selection.

> Response: Since we wanted to assess how generalized our findings are, we chose two RPSs that are close to RPS3 (RPS19 and RPS20) as well as RPS6 far away from RPS3. Similarly, two RPLs in close proximity (RPL5 and RPL11) as well as one distant one (RPL7) were chosen. Some of our choices were also guided by the fact that individual ribosomal proteins are known to be mutated in ribosomopathies and/or in cancer. An overview for the ribosomal proteins targeted in our study is shown below (**Reviewer Table 1**).

Ribosomal protein	New name	Location within the rib. particle	Role in ribosomopathies	Role in cancer
RPS3	uS3	beak	/	/
RPS6	eS6	right foot, far away from RPS3	/	/
RPS19	eS19	head, neighbor of RPS20	DBA	Epidermoid carcinoma
RPS20	uS10	head, direct neighbor of RPS3	DBA	Chronic lymphoblastic leukemia, colorectal and gastric cancer, endometrial carcinoma
RPL5	uL18	central protuberance, neighbor of RPL11	DBA	T-ALL, melanoma, multiple myeloma, glioblastoma, breast cancer
RPL7	uL30	close to the P stalk	/	/
RPL11	uL5	central protuberance, neighbor of RPL5	DBA	T-ALL, melanoma, gastric cancer

Reviewer Table 1 – Criteria for RPSs and RPLs used in this work. No role reported (/); Diamond-Blackfan-Anemia (DBA); T-cell acute lymphoblastic leukemia (T-ALL). Data taken from the following review articles: Kang et al, 2021, PMID: 3446242; Farley-Barnes et al, 2019, PMID: 31376929; Vlachos, 2017, PMID: 29222326.

ZNF598 silencing does not seem to be as efficient as the one for RNF10. It might be helpful to quantify figures such as S1e, where there seems to be difference between Ub1 levels. Do authors think there might be any residual activity?

> Response: When looking at **Fig. 1a**, KD efficiencies of RNF10 and ZNF598 appear to be similar. To improve the KD of ZNF598, we now performed a double KD of ZNF598 over a period of 2 days, and observed the same: no reduction in Ub₁-RPS3 levels in the ZNF598 KD HeLa cells (new **Suppl. Fig. 1i**).

This result is somewhat in contrast to Garzia et al (2021, PMID: 34348161), who showed that Ub₁-RPS3 levels were decreased in ZNF598-KO HEK293 cells, and absent in RNF10-KO HEK293 cells, upon elongation stalling induced by ANI treatment. In contrast, Ub₁-RPS3 levels were unaffected in ZNF598-KO HEK293 cells (but again absent in RNF10-KO HEK293 cells) upon treatment with the translation initiation inhibitors Harringtonine and Hippuristanol.

At this point, we can only speculate that this discrepancy might be due to differences between HEK293 and HeLa cells. Alternatively, the KO of ZNF598 (in contrast to acute KD) might lead to

adaptation, e.g. altered expression levels of RNF10. In fact, in Figure 5 of Garzia et al (2021, PMID: 34348161), there is some evidence for this. We would like to emphasize that our observation is concurrent with the central findings of Garzia et al (2021) and Garshott et al (2021), i.e. that RNF10 is the main E3 ligase responsible for RPS3 monoubiquitination.

Additionally, it is interesting that the RNF10 levels in Fig. 1a (lanes 1,2,5,6) and S1b (lanes 1,2,3,4) seem to be different in relation to the control. Considering that the same siRNA S193 was used, can authors explain this? Is there any possibility in any way that ZNF598 influences RNF10 levels?

> Response: This comment now pertains to **Fig. 1a** and **Suppl. Fig. S1c**. The Western blot we showed in former Suppl. Fig. 1b was not representative due to loading issues. After going through all our repeats, we became aware that RNF10 increases upon ZNF598 KD in RPE1 cells. Therefore, a new blot is now shown in **Suppl. Fig. 1c**, and the quantification of RNF10 levels in RPE1 cells was added (new **Suppl. Fig. 1d**). The increase in RNF10 is approximately 1.5-fold upon ZNF598 KD in RPE1 cells.

Fig. 1a shows the KD of ZNF598 in HeLa cells, where we do not observe a change in RNF10 expression under basal conditions. We now added a quantification of RNF10 levels in HeLa cells (new **Suppl. Fig. 1a**). The effect of ZNF598 KD on RNF10 levels under conditions of RPS3 KD is addressed above in Reviewer Fig. 3.

In the relation to the line 111, in the Fig. 1c authors comment that there is no monoUb in the fraction #1. It appears that there is a weak band which seems very minor. However, it is interesting that the Ub1 levels seem to fluctuate with the same trend as RPS3 levels. It would be helpful to compare these ratios by quantification and accordingly adjust narrative if needed.

> Response: We now changed the experiment that we show in Fig. 1c by using pools of different RNF10-KO clones instead of two single RNF10-KO clones, with the aim to avoid clonal effects (see response to reviewer 1). As the reviewer noticed correctly, there is a slight Ub₁-RPS3 band in the first fraction of both parental HeLa and the KO control upon ANI treatment. We now quantified the Ub₁-RPS3 / RPS3 ratio across the polysome profiles (new **Fig. 1d**), which shows that Ub₁-RPS3 levels do not simply fluctuate with the RPS3 levels. Rather, monoubiquitination of RPS3 is strongly enriched in fractions 2-5, corresponding to the 40S, 60S, 80S and disome fractions.

In fact, there is an interesting difference in the distribution of Ub₁-RPS3 between ANI-treated cells and cells subjected to RPL7 KD, which we now show as a direct comparison in new **Fig. 5g** and new **Suppl. Fig. 5d**. While RPS3 monoubiquitination is restricted to the free protein and 40S fraction (#1 and 2) upon RPL7 KD, it extends much farther into the polysomal fractions upon ANI treatment. This is well compatible with RPS3 monoubiquitination being triggered by initiation stalling (RPL KD) versus elongation stalling (ANI).

Minor comments

Line 62

Further comment might be needed to avoid confusion as Hel2 in humans is a 14-3-3 type protein.

> Response: We now wrote "yeast Hel2" to make sure that there is no confusion with the human HEL2 protein.

Line 65

Was RNF10 the only outstanding candidate? If no, it might be helpful to provide an alignment figure and logic why only RNF10 was tested.

> Response: When we started the project in 2019, the human homologue of yeast Mag2 was still unknown. Thus, we aligned the Mag2 amino acid sequence (accession number Q06436) to the proteome of humans and other metazoan organisms using the protein blast tool from NCBI (<https://blast.ncbi.nlm.nih.gov/Blast.cgi>). The result showed the best alignment of Mag2 with the RING finger protein 10 (RNF10) of humans and other metazoa, with an E value much smaller than the one for the next best hits (**Reviewer Table 2**). Therefore, we started to test whether RNF10 is a functional homolog of yeast Mag2. Since loss of RPS3 monoubiquitination was evident in our first RNF10 KD experiments, we did not consider other candidates.

Common name	Scientific name	Description	Query coverage	E value
Human	Homo sapiens	RING finger protein 10	28 %	10 ⁻¹³
Human	Homo sapiens	tripartite motif-containing 68	9 %	10 ⁻³
Human	Homo sapiens	E3 ubiquitin-protein ligase TRIM68	9 %	10 ⁻²
Human	Homo sapiens	Ro/SSA1 related protein FLJ10369	9 %	10 ⁻²
Mouse	Mus musculus	RING finger protein 10	28 %	10 ⁻¹²
Zebrafish	Danio rerio	RING finger protein 10	39 %	10 ⁻¹⁴
Chicken	Gallus gallus	RING finger protein 10	40 %	10 ⁻¹³
Common frog	Rana temporaria	RING finger protein 10	63 %	10 ⁻²⁵

Reviewer Table 2 – Blast search for homologs of yeast Mag2. The yeast Mag2 amino acid sequence was aligned to the proteome of humans and other metazoa by protein blast (<https://blast.ncbi.nlm.nih.gov/Blast.cgi>); shown are the top scoring results.

Line 184

RIO2 is introduced here without any previous context. Can authors explain why they probed this protein in particular?

> Response: In our initial submission, we chose to knock down RIO2, an essential factor in late-stage pre-40S maturation (Zemp et al, 2009, PMID: 19564402), to perturb ribosome biogenesis through other means than RPS or RPL KD. However, since the effect of RIO2 KD on RNF10 expression is rather mild (compared to the strong effect of RPS KD), we now decided to take out the RIO2 data from our revised manuscript. Thereby, we can streamline our story and make space for the new data.

Line 237

Considering that authors performed quantification in Fig. 5b by RNF10 ratio to Tubulin, I wonder whether the measured values for Tubulin in Fig. 5a remained in the same range. Visually it appears that these levels fluctuate.

> Response: This comment now refers to **Fig. 6a and b**. We agree that according to the Tubulin staining, loading in **Fig. 6a** is not perfect, though it is sufficient for normalization. In **Reviewer Figure 8a and b**, the reviewer can see the other two biological replicates of this experiment, where the Tubulin staining is more even. However, since some of the other stainings are not ideal in these repeats (in particular RPS20), we decided to keep the blot in **Fig. 6a**. The effect of RPS3, RPS6, RPS20 and RPL7 KD on RNF10 levels is clearly visible in all repeat experiments.

Lines 240 & 513

Fig 5.d includes western blot for p21. It would be helpful to explain this in the text.

> Response: This comment now refers to **Fig. 6**. p21 is a major target gene of p53, and thus serves as a read-out for p53 activity. Since we removed panel d of former Fig. 5 from our manuscript (in response to a comment of reviewer 1), an explanation of p21 is no longer needed.

Lanes 472 & 483

Although the rotor type is mentioned here, it would be helpful to unify the units used during centrifugation to RCF.

> Response: We used a swinging bucket rotor. Thus, g is different at the top and at the bottom of the tubes. $RCF_{max} = 208600 \times g$; $RCF_{average} = 157600 \times g$. Therefore, we originally mentioned only the rotor type and RPM. We now added RCF_{max} and $RCF_{average}$ to the methods description.

Response to reviewers:

Reviewer #1 (Remarks to the Author):

The authors have done a thorough job addressing reviewer comments and I am now enthusiastic about publication. Below are listed a few lingering questions that are NOT essential to address for publication.

We thank this reviewer for expressing his/her enthusiasm for our study.

1. The revised manuscript more thoroughly cites published work that demonstrated RNF10-mediated ubiquitination of RPS3 in response to elongation/initiation inhibitors.

- Garzia et al 2021 and Garshott et al 2021 should be cited in the first mention of Fig. 1a. I appreciate that the authors have demonstrated RPS3 ubiquitination in response to a wider variety of elongation inhibitors, but both these papers previously showed the same result with anisomycin and should be acknowledged alongside the data presented here.

> Response: We now added these two citations at the first mentioning of Fig. 1a (line 87).

2. The authors have made efforts to test phenotypes associated with RNF10 KO. They find that RNF10 KO causes minor increases in halfmer formation with low dose anisomycin treatment. RNF10 KO cells also have slightly impaired proliferation under basal conditions that becomes more pronounced under low dose anisomycin.

- It would be useful to show the absolute proliferation rates for untreated and anisomycin-treated cells.

> Response: We changed the y-axis on the left graph in Fig. 2g (former Fig. 2f) so that it now shows the absolute proliferation rate b derived from the equation $n = 2^{(b \times t)}$; t being time and n being the cell number; the unit of b is [1/h].

- Could the authors comment on why protein synthesis (as measured in their puromycin incorporation assay) does not appear to decrease with anisomycin treatment, despite anisomycin significantly impairing proliferation? Further, it seems odd that RNF10 expression affects proliferation but not protein synthesis – could the authors comment on this?

> Response: We added a titration experiment (Fig. 2f based on quantification of Suppl. Fig. 2g and replicates thereof) showing the effect of increasing concentrations of anisomycin (ANI) on puromycin incorporation as a measure of total protein synthesis, and histone H3 serine 10 phosphorylation as a measure of mitotic cells and hence proliferation. The experiment shows that ANI at 0.02 $\mu\text{g/ml}$ causes only a mild reduction in total protein synthesis (by about 20%), while the proliferation signal has already dropped by >80%. Based on this dose response curve, we had chosen to conduct our proliferation experiments with chronic ANI exposure at 0.02 $\mu\text{g/ml}$. This result illustrates quite impressively that cells respond to mild perturbations of protein synthesis by actively downregulating their proliferation rate, reflecting activation of the ribotoxic stress response.

The reviewer also noted correctly that KO of RNF10 does not seem to have any strong effect on total protein synthesis. Interestingly, re-introduction of RNF10-WT into HeLa RNF10-KO cells, which leads to RNF10-WT overexpression, antagonizes the drop in polysomes that is normally seen upon ANI treatment (quantification of polysomes in Suppl. Fig. 2a). Hence, our data would argue that RNF10 induction becomes important for cells under conditions of perturbed protein synthesis, and that its activity is connected to the ribotoxic stress response.

3. The authors have made some effort to explore RNF10 degradation using the proteasome inhibitor MRZ.

- They propose that smearing above the RNF10 band upon MRZ treatment may reflect ubiquitination of RNF10. To test this, they could IP their tagged RNF10 constructs and probe them using a ubiquitin antibody.

> Response: We agree that such experiments would help to pursue the mechanism by which RNF10 degradation is induced. We would like to point out that the IP in such an approach has to be extremely clean and is typically conducted under denaturing conditions, in order to ensure that any band seen with the Ub-antibody directly reports on ubiquitinated RNF10 and not any other associated or contaminating protein in the IP. This experiment will require extensive optimization and is thus beyond the scope of our revision.

- The authors comment that MRZ treatment is toxic to cells after 8 hours and prevents RPS3 ubiquitination. Is MRZ treatment inhibiting translation? This could be tested with polysome profiles of MRZ-treated cells and may explain why RPS3 is not ubiquitinated if translation initiation is inhibited by MRZ.

> Response: At this point we would abstain from pursuing the mechanism that is responsible for the negative effect of MRZ treatment on RPS3 ubiquitination. The reason may well be connected to an effect of MRZ on translation as suggested by the reviewer, yet proteasome inhibition is known to have pleiotropic effects on almost every process in the cell, making it very difficult to untangle direct from indirect effects.

- The MRZ data argue that further stabilization of RNF10 does not explain increased RNF10 levels with RPL knockdown. The authors could test whether translation of RNF10 is increased upon RPL knockdown by checking whether the amount of RNF10 mRNA in polysomes increases (or not) upon RPL knockdown.

> Response: We agree that this experiment would help to understand the mechanism by which perturbation of 60S biogenesis augments RNF10 levels. However, from our own experience we know that these experiments are very tedious and need to be conducted with utmost care (Hisaoka et al, RNA Biol 2022;19:437-452, PMID: 35388737). Rather than including preliminary data on this aspect in our revised manuscript, we prefer to explore the principles underlying RNF10 regulation (both the degradation upon RPS KD and the induction upon PRL KD) in a separate study and with sufficient time to explore potential mechanisms in detail.

Reviewer #2 (Remarks to the Author):

Review for NCOMMS-22-43495A

I appreciate the author's efforts to prove their proposal. However, I want to highlight that in the revised manuscript, there are still several overstatements without direct evidence to support the claim of 'the dissociation,' such as "The E3 ubiquitin ligase RNF10 promotes dissociation of stalled ribosomes" in the title and "RPS3 ubiquitination at K214 promotes dissociation of stalled 40S subunits" in Figure 3 legend. It's challenging for me to recommend the publication without the correction of these overstatements.

> Response: We have addressed this concern extensively in our first revision (see response to the 2nd reviewer's 2nd major point in our first rebuttal letter), including several new experiments that were added during the first revision (Suppl. Fig. 2f, Fig. 2c-e, Fig. 3b and c). All the results we obtained support our hypothesis that the observed suppression of ribosomal half-mers by RNF10 should be interpreted as RNF10 promoting the dissociation of stalled 40S subunits. In contrast, we have no evidence for the alternative hypothesis, i.e. that RNF10 inhibits dissociation of 60S subunits from stalled ribosomes, which is triggered by the RQT/ASC-1 complex. This notion is further supported by a recent publication from the Pestova lab (Miscicka et al., NAR 2024, reference 28 in our manuscript), showing by *in vitro* reconstitution of stalled ribosome complexes that RNF10 does not affect the activity of the mammalian RQT/ASC-1 complex responsible for 60S dissociation. We discuss this aspect carefully in our manuscript by phrasing both hypotheses, and we now improved the wording of our arguments in this passage (line 164-182). Based on the evidence we collected

and the recent publication from the Pestova lab, we feel it is safe to posit that RNF10 promotes the dissociation of stalled 40S subunits.

On a general note, we would like to point out that scientific progress is always based on the interpretation of data, and that it is good practice to move forward with the most plausible and reasonable model as long as it is supported by experimental evidence. Remaining purely descriptive does not help to conceptualize data, which is at the heart of the scientific process.

Reviewer #4 (Remarks to the Author):

Lehmann et al. submitted a revised manuscript characterizing E3 ubiquitin ligase RNF10 and its relation to the stalled ribosomes during translation. Additional experiments helped to clarify some of the conclusions and link the related sections. I appreciated the addition of the final diagram and also the figures in the comments to the reviewers. Some of these, as well as the tables, would be helpful for the main text too. Several additional comments have been made below.

General Comments

I found the magnification of the polysome profiling very helpful. I would suggest to include more of them, especially in Fig. S2bc, S5cd. Additionally, some of the profiles exhibit dips in the absorbance (Fig. 1 or Fig. S5). It would be good to comment on this observation. Lastly, the authors mention free fraction of the profiles in the text. It would be good to mark this on the X-axis.

> Response: We now added enlarged views of the polysome curves in Fig. 7c, Suppl. Fig. 2b and c, and Suppl. Fig. 5d.

The dips at regular intervals in the polysome profiles of Fig. 1c and Suppl. Fig. 5d are due to an electric signal generated by the fractionator, and hence represent artefacts. We referred to this problem in both Figure legends with the following sentence: "Sharp peaks pointing downwards are artefacts of the electric signal of the fractionator."

We also added the label "free" to the left part of the polysome profiles in Figures 1c, 2b, 2c, 3b, 4a, 5a, 7a and 7c, as well as in Suppl. Figures 2b, 2c, 2d, 3, 4g, 4h, 5c, 5d and 7a.

In the experiments presented in Fig. 4 and 5, multiple RPS and RPL were silenced. This appears to have a sequential effect where the silencing of one protein leads to reduced levels of the other analysed proteins in the subunit. It would be very helpful to provide some additional comments or accompany these blots with quantification. Does this silencing trigger a cascade effect where the whole subunit is being cleared? Additionally, the silencing of the 40S proteins effect on 60S levels visually appears to be stronger than vice versa. I wonder if the data can be interpreted and thus contribute to the conclusions and model authors presented.

> Response: We agree with the reviewer's observation that in Fig. 4a and 5a, KD of small ribosomal proteins has a stronger effect on 60S subunit accumulation than KD of large ribosomal proteins on the accumulation of 40S subunits. On the other hand, the effect of RPL KD on 40S accumulation was much stronger in Fig. 7a. Moreover, the changes in 40S accumulation appear to be quite dynamic with maximum accumulation observed after 28 hours of RPL7 KD (Fig. 7c). We would caution the reviewer not to overinterpret these differences since they might have technical reasons, e.g. variations in the efficiency of KD between repeat experiments.

Generally speaking, 60S/40S accumulation occurs in response to the generation of defective 40S/60S subunits upon RPS/RPL KD. According to data from our and many other labs, the defective subunits are primarily detected during failed attempts of translation initiation and elongation. Subsequently, these defective subunits are degraded through pathways that (in the case of 40S) involve RNF10 and monoubiquitination of RPS3 at K214. Since KD of a single RPS/RPL protein causes degradation of the entire 40S/60S subunit, it is not surprising that knocking down one ribosomal protein causes reduced levels of the other proteins of the same subunit. Whether there are interdependencies between specific ribosomal proteins beyond this basic quality control mechanism is outside of the scope of our study.

Major comments

Authors have defined RQC as a process that includes the degradation of the nascent peptide, ribosomal proteins, and affected mRNA. Based on this definition, if the ribosomes stop due to the mRNA-related effects, efficient RQC should result in the clearance of these elements and restart of the translation.

Considering that the polyLysine reporter induced stalling, but some mCherry signal is still observable, a percentage of the translation runs into completion. Wouldn't this suggest that KD of ZNF598 results in higher clearance of the stalled components and thus re-enabling the system to carry out translation and thus increase mCherry production? If so, then the statement in Line 172 about reduced RQC efficiency seems confusing. I suggest defining RQC efficiency and rewording this narrative to make the characterized process easier to understand. Additionally, explaining the link between RQC and RQT would be helpful. Similar could be done for ZNF598 and RNF10 to make the hypothesis clearer. Overall, this section (Lines 164-190) introduces many elements with some logical links missing.

> Response: RQC is the process by which stalled and collided ribosomes are recognized, leading to the dissociation of the ribosomal subunits from the mRNA and the degradation of the nascent polypeptide. ZNF598 acts very early in RQC by recognizing the 40S interface of collided ribosomes and ubiquitinating uS10 (= RPS20 in human and yeast), and is required for initiating RQC. Accordingly, KD of ZNF598 represses RQC, allowing for some read-through on the stalling site of the reporter mRNA, which leads to elevated levels of mCherry and is reflected by an increase in the mCherry / GFP ratio. This has been described in the original publication of the reporter assay (Juzskiewicz and Hedge, 2017, Mol Cell 65:743-750; reference #37 in our manuscript), and is also observed by us in Suppl. Fig. 2e and f. To document the FACS gating strategy of the RQC reporter gene analysis, we now added panel e to the Suppl. Fig. 2. Thus, our statement on line 173-175 is correct: "As expected, KD of ZNF598 specifically reduces RQC efficiency on the poly-lysine (K^{AAA})₂₀-containing reporter mRNA that imposes ribosome stalling (Suppl. Fig. 2e and f, visible as elevated mCherry / GFP ratio)".

The ribosome quality control trigger complex (RQT) acts downstream of ZNF598 in RQC by inducing the splitting and dissociation of collided ribosomal subunits. Specifically, RQT promotes dissociation of the 60S subunit from the leading ribosome of a stalled disome (Best et al, 2023, reference 14 of our manuscript). We have now changed our wording in the text to better explain the role of ZNF598 and RQT in RQC, and improved the interpretation of the RQC reporter assay (line 164-182).

Lines 293-296/600 and Fig. S5f – This passage is somewhat confusing. The authors show the figure with the Ub-K48 label, however, the K48 linkage-specific antibody is not mentioned in the methods. Does this panel represent polyubiquitinated RPS3? This is also not described in the text.

Furthermore, do authors by recognition mean proteasomal recognition? Canonically, K48-linked polyubiquitin chains of four or more ubiquitin molecules are required for efficient proteasomal degradation, although degradation of monoubiquitylated targets is also possible. This section of the text needs to clarify whether the observed event is related to degradation or just recycling, and the type of modification that was observed.

> Response: Indeed, we forgot to mention the Ub-K48 antibody, which we now added to the antibodies listed in the methods section (rabbit anti-ubiquitin-K48, Abcam, ab140601). Suppl. Fig. 5f shows total Ub-K48 linked proteins present in the protein lysate and does not represent ubiquitination of any specific protein. We used this antibody as a control to show that MRZ efficiently inhibits the proteasome. This is now explained better in the text (line 298-300), and "total" was added to the labelling of the Western blot in Suppl. Fig. 5f.

Minor comments

Fig. S1g – KO-control is not described. Does this represent KO by CRISPR/Cas9 without gRNA?

> Response: The KO-control clone underwent the same CRISPR/Cas9 procedure but without gRNA, and tested positive for RNF10 (sequencing and Western blot analysis). We now added its description to the methods section in the manuscript (line 526-527): "The HeLa KO-control clone underwent the same CRISPR/Cas9 procedure but without gRNA". To improve consistency of our nomenclature, we also harmonized the labelling of the KO-control clone in Fig. 1b as well as in Suppl. Fig. 1f and 1g.

Line 103 – Authors claim that ZNF598 KD had no effect on Ub1-RPS3 levels, however, siRNA 201 coupled with ANI and CHX treatment appear to have lower band intensity. Quantification would help to make this clearer.

> Response: In most experiments, we do not observe changes in RPS3 monoubiquitination upon ZNF598 KD. The slight reduction observed by the reviewer in Suppl. Fig. 1h is only present with siRNA 201, and not with siRNA 193 (which induces a better knockdown of ZNF598). Since the focus of our manuscript is on RNF10 and not ZNF598, we do not wish to pursue such minor differences that we do not observe consistently. We now chose to phrase this passage more carefully (line 103-104): "...yet KD of ZNF598 in our hands had no consistent effect on RPS3 monoubiquitination in HeLa cells subjected to the translation elongation inhibitors ANI, CHX or BLA (Fig. 1a, Suppl. Fig. 1h and i)."

Line 172/621/622/Fig. S2e – reporter naming. It would be good to unify nomenclature. I would also suggest omitting AAA usage as the mixing of amino acid and nucleotide codes is confusing and suggests a Lysine/Alanine sequence.

> Response: This reporter gene was originally described by Juszkievicz and Hedge (2017, Mol Cell 65:743-750; reference #37 in our manuscript), who used the nomenclature (K^{AAA})₂₀ throughout their publication. To remain consistent with the original publication, we prefer to use the same nomenclature, both in our text and in Suppl. Fig. 2e and f.

Fig. 3a – K214A shows a minor Ub1 band upon ANI treatment which is comparable with the WT in the RPS3 panel but not in the FLAG panel. Can the authors please explain this discrepancy? Considering this experiment is done pre-silencing, do authors think that there might be a lower efficiency of FLAG-RPS3 incorporation into the ribosomal subunit, thus generating two populations? One way of testing this would be to show Ub1-RPS3 levels from the profiling fractions and compare free vs. 40S levels, similar to Fig. 1c

> Response: The minor Ub-RPS3 band in RPS3-K214A cells upon ANI treatment results from residual endogenous RPS3. These cells were not transfected with siRNA against RPS3. They stably express FLAG-RPS3-WT or FLAG-RPS3-K214A protein, which replaces most, but not all, of the endogenous RPS3 protein in ribosomes.

Line 244 – How were the RPS20 levels of 50% calculated/quantified?

> Response: This was judged by eye from the Western blot in Fig. 4d. We now added a proper quantification of the RNF10, RPS3 and RPS20 protein signals upon RPS3 and RPS20 KD in a new panel (Fig. 4e). We adjusted the statement in our text (line 251) since RPS20 drops to about 50% already after 18 hours of KD.

Line 287 – figure number missing

> Response: We added the figure number to the manuscript (now line 294) and thank the reviewer for so carefully going through our manuscript.